# Synthetic Data Generation for Deep Learning-Based Inversion for Velocity Model Building

Apostolos Parasyris * , Lina Stankovic and Vladimir Stankovic

Department of Electronic and Electrical Engineering, University of Strathclyde, Glasgow G1 1XW, UK; lina.stankovic@strath.ac.uk (L.S.); vladimir.stankovic@strath.ac.uk (V.S.)
* Correspondence: apostolos.parasyris@strath.ac.uk

**Abstract:** Recent years have seen deep learning (DL) architectures being leveraged for learning the nonlinear relationships across the parameters in seismic inversion problems in order to better analyse the subsurface, such as improved velocity model building (VMB). In this study, we focus on deep-learning-based inversion (DLI) for velocity model building, leveraging on a conditional generative adversarial network (PIX2PIX) with ResNet-9 as generator, as well as a comprehensive mathematical methodology for generating samples of multi-stratified heterogeneous velocity models for training the DLI architecture. We demonstrate that the proposed architecture can achieve state-of-the-art performance in reconstructing velocity models using only one seismic shot, thus reducing cost and computational complexity. We also demonstrate that the proposed solution is generalisable across linear multi-layer models, curved or folded structures, structures with salt bodies as well as higher-resolution structures built from geological images through quantitative and qualitative evaluation.

**Keywords:** full waveform inversion (FWI); deep learning-based inversion (DLI); velocity model building (VMB); deep neural networks (DNNs); seismic inversion

## 1. Introduction

Seismic inversion plays a decisive role in the study of the underground structure of Earth, be it investigations at the near surface or greater depths. Subsurface exploration over a large area is traditionally achieved through the generation of artificial energy sources on the ground surface, within boreholes, or through a combination of the two, in parallel with placed sensors that record the generated seismic waves that travel through the Earth (i.e., seismic data). Surface seismic surveys, due to their relatively low cost, are often adopted for near-surface exploration (depths < 1 km). Geophysical methods, in general, are used for site characterisation, hydrocarbon exploration, and analysis of natural earthquakes or human-induced seismicity [1], leading to association with many branches of structural [2–4], industrial [5–7] and seismological analysis or research [8].

The aim of seismic methods is to reveal unknown physical properties of the subsurface geomaterial, such as density, shear and compressive velocity, by analysing the propagation of seismic waves (reflected, refracted or surface waves) through the Earth. Seismic methods for Earth modelling, like most geophysical methods, have as their main objective, to derive both quantitative and qualitative conclusions about the structure of the Earth through the use of sensor measurements. This structure is represented by physical quantities, such as density $\rho$, longitudinal wave velocity $V_p$ and transverse wave velocity $V_s$, which are usually described by a finite number of parameters or by a finite set of functions [9]. Often, only a 2D model of one or a few of these attributes is constructed, with the acoustic longitudinal wave velocity model building (VMB) being the most popular [10]. VMB is critical for the characterisation of subsurface material, i.e., each material has its own $V_p$ ranging from 0.40–2.50 km/s in ordinary soils (sand, clay) to greater than 2.00 km/s for rocky soils [11]. VMB is also important for seismic and microseismic event analysis and specifically for

source-localisation algorithms [12]. Furthermore, precise prediction of earthquake source parameters is one of the most significant tasks in modern seismology for analysing fault systems and tectonic processes [8,13].

A widely adopted approach for geophysical inversion is full waveform inversion (FWI) [14], as it enables high-resolution imaging in complex sets [15,16] that can be used in multiple applications and, unlike conventional seismic methods that use only a small part of the wavefield, takes advantage of the full content of the seismic record. The key challenge of FWI is that the parameters that compose the forward physical problem are complex and non-linear, which means that the solution to the inverse problem cannot be approached analytically and needs to be solved with iterative approximations. Despite the complex mathematics behind FWI, of which VMB is a subset, FWI offers the most accurate way of creating velocity images of the subsurface. FWI [17,18] aims, through iterative steps, to perform non-linear inversion to reveal a subsurface model by minimising the difference between recorded data that were acquired in a seismic survey (real measured data recorded by placed equipment/recorded shots) and synthetic data (shots which are calculated for an adopted hypothetical model of the subsurface). In each iteration, the initially adopted Earth model is adjusted and produces new synthetic data until the difference between the recorded and synthetic shots is minimised. This iterative calculation takes place if the velocity image of the subsurface is required to be produced from known recorded shots. If the opposite is needed (given the velocity model to define the wavefield), the procedure is a forward solution of the wave equation and is thus not iterative. Figure 1 summarises the forward and inverse problems. The prediction of the Earth model is limited by the initial choice of the model at the start of the iterative process. Additionally, current seismic sensors are usually high pass, limited to above 3 Hz, and therefore, recordings suffer from lack of low-frequency measurements [19]. This makes it harder to achieve convergence to a local minimum due to the cycle skipping problem, and in combination with the strong dependence of the method on a successful starting velocity model, it degrades the effectiveness of the method.

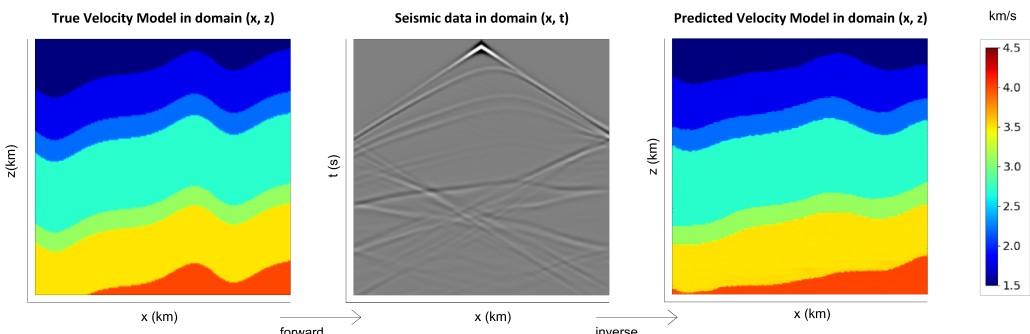

**Figure 1.** True 2D velocity model (**left**); seismic shot (**middle**); predicted 2D velocity model (**right**). The forward problem leads to modelled seismic data, by solving the wave equation when the velocity model is given, while the inverse seismic problem leads to a predicted velocity model when the recorded shot is given. The major advantage behind the deep learning solution for the inverse problem is that the networks can learn the non-linear relationship of the parameters by being trained on pairs created from the forward solution.

FWI solutions were introduced in the 1980s [20,21], but major improvements in computational performance have highlighted FWI and have made it more feasible as a computational tool [22], even for 3D reconstruction [23,24]. Furthermore, the high computational performance provided by the graphical computing units (GPUs) that have taken place in the last few years have provided the ability to perform seismic inversion with advanced machine learning (i.e., deep learning) on large and highly complex datasets, gaining significant benefits over physics-based FWI. The major advantage is that the deep learning (DL) networks provide quite stable predictions for the velocity model, while they can learn the non-linear relationships by being trained on pairs obtained from the forward solution of

the wave equation, which is relatively easy to solve. After the training process, a nonlinear map, relating the system parameters, is created, which is then used for solving the inverse problem. In DL-based inversion (DLI) methods, it is possible to learn the low-frequency content from simulated data or from prior information data to thus overcome the obstacle of missing frequencies in the seismic recording, which is the main problem in the practical application of seismic inversion [25]. In parallel, various studies [25,26] have shown that DLI models can demonstrate better performance compared to FWI solutions. However, DLI requires training on thousands of representative velocity models, which are significantly difficult to generate. At the same time, in the current literature, it is hard to find methods for the massive generation of pragmatic training models or available code as well as for the evaluation of the performance of DLI architectures for data resulting from a variety of construction techniques for the models. For example, the current literature is missing evaluations of performance of examined DLI architectures simultaneously for horizontally homogeneous, folded, and salt datasets together with models created directly from geo-logical images presenting rich information- granularity or models that take advantage of geometries seen in real geological layers, all integrated within a mathematical framework.

Although FWI is the most effective approach for reconstructing strongly heterogeneous subsurface velocity models, for multi-source modelling, FWI is computationally intensive, while DLI also requires the wavefield of each model to be derived from a separate forward solution, in turn increasing the computational cost of training deep networks since it has to be repeated thousands of times. With regard to the selected number of shots, which is an absolutely decisive parameter of vertical cost increase, studies have shown that only a few shots are required for constraining the acoustic $V_P$ model very well, especially for the case of near-surface FWI [27]. Other studies with the aim of reducing this computational cost have examined new methods that have been extended beyond conventional FWI by simulating super-shots with the use of simultaneous encoded sources through the linear combination of all simulated shots. In other words, simultaneous sources are activated at the same time along a geological section in order to be summed and to create super-shot gathers, thus reducing the required number of individual simulations. The problem in this case is that simultaneous sources introduce cross-talk artifact noise in the final image due to interference among the individual sources within a super-shot, and thus, attempts have been made in order to balance this disadvantage either by assigning random weights to the source wavefields [28], or by resampling source positions in every iteration during the computation process [29]. Unlike FWI, DLI methods do not require any starting models or many simulating sources to accurately perform seismic inversion, and they are less sensitive to reflections from the borders of the physical models, which usually lead to poor FWI results that are less sensitive to missing low frequencies [25] and are not prone to the cycle-skipping problem [10]. Once the DLI model is trained, then the estimation runs in just a few seconds.

Therefore, the two key challenges in the adoption and evaluation of DLI solutions are a standardised approach to generate representative Earth models, for training and testing and reducing the need for multiple-sourced simulation, and hence computational complexity, for accurate subsurface reconstruction. In this study, in order to address the aforementioned challenges and more specifically in order to ensure the high performance of the network for the minimum amount of possible seismic sources and to provide a clear velocity modelling methodology, we introduce a complete mathematical framework for constructing representative velocity models and then use it for the training and testing of two distinct DL architectures: the PIX2PIX conditional generative adversarial network, originally proposed for image-to-image translation [30], and the fully convolutional velocity model building (FCNVMB), proposed for VMB [25]. For the sake of completeness, we include the prediction of the acoustic velocity $V_p$ model via FWI. The value of a robust approach to generate training and test sets is demonstrated via the DLI architecture performance. The results highlight that PIX2PIX consistently provides a stable solution with variations in datasets unlike FCNVMB, indicating PiX2PIX's superior replicability performance. Additionally,

computational complexity is evaluated for consistency of performance using one source. Again, PIX2PIX provided consistent performance across a range of datasets with one source, unlike FCNVMB. The contributions of this paper can be summarised as follows:

- We define a multi-component and easy-to-use mathematical approach for generating representative samples of multi-stratified velocity models that can be used to train DLI for VMB models, resulting in linear models and models with folds (Section 3.1) as well as models with salt bodies (Section 3.2).
- We define an autonomous method for generating velocity models directly through processing pixel intensities of geological images that present stratified layers with application of elastic displacements (Section 3.3).
- We propose a PIX2PIX conditional generative adversarial network (cGAN)-based architecture for DLI, using the ResNet-9 architecture as a generator that achieves stable solutions, with low variability in performance, when simulating only a single shot per model (Section 3.4).
- Demonstration of the generalisability of the proposed solution across all proposed velocity models via the commonly used peak signal-to-noise ratio (PSNR) and structural similarity index (SSIM) performance metrics as well as visual quality of the reconstructed velocity images, noting clear demarcation across stratifications and minimising artifacts (Section 4)
- Benchmarking against FWI, PIX2PIX with the original U-Net architecture and FCN-VMB of [25], also based on the U-Net architecture, with a single shot (Section 4).
- Rigorous evaluation of the model across all proposed velocity models, both qualitatively and quantitatively (Section 4).
- Release of a comprehensive dataset (DOI: https://doi.org/10.15129/d49bcfc6-7bd0-450c-9734-cf89403ef9c0, accessed on 12 April 2023) of a velocity model and wavefield pairs for all five proposed multi-stratified velocity models, comprising linear, folded, salt bodies and direct Earth-model-generated images.

This paper is organised as follows. We first provide a detailed literature review on data-driven VMB in Section 2, to position our contributions with respect to the state of the art. This is followed by the proposed methodology in Section 3, both for generating representative samples of multi-stratified velocity models with elastic displacements as well as for adapting PIX2PIX GAN and FCNVMB architectures for 1D and 2D heterogeneous acoustic VMB. The performance of the proposed methodology for VMB, benchmarked on a minimum number of seismic sources, is presented in Section 4, before we conclude the paper.

## 2. Deep-Learning-Based Inversion for Velocity Model Building

This section first reviews DLI approaches, focusing on those developed for solving the VMB problem, which have emerged recently. In particular, we identify the types of DL architectures attempted, the types of datasets on which performance has been evaluated and the number of shots used for each simulation. Then, a review of the approaches used by these architectures to synthesise velocity models is provided.

### 2.1. DL Approaches for VMB

A comparative study for VMB between a convolutional neural network (CNN), a recurrent neural network (RNN), a long short-term memory (LSTM) and a gated recurrent unit (GRU), was presented in [10]. A total of 12,000 velocity models were used with 4–8 background subsurface layers in velocity intervals of 2.00–4.00 km/s out of which 9000 contained salt bodies in velocity intervals of 4.45–4.55 km/s with varying shape and position. The associated seismic gather was produced for 3 shots and 51 receivers, while the data were split with 9600 velocity models for training and 2400 for testing. The results of this comparison showed that the CNN architecture required 7,182,728 learnable parameters, which was 4.61 times more than the RNN architecture, 2.50 times more than the GRU architecture and 2.03 times more than the LSTM architecture. The RNN was the least

complex in terms of parameters, but the GRU and LSTM models demonstrated the best performance with a PSNR of 30.58 dB and SSIM of 0.8515.

A fully convolutional network, named FCNVMB, uses an encoder–decoder architecture for acoustic VMB directly from raw seismic data [25]. The encoder adopts a modified U-Net architecture, synthesised by ten 2D convolutional layers, while the decoder is structured by eight 2D convolutional layers connected with the corresponding deconvolutional layers. Two types of velocity datasets are used, both containing salt bodies. The first simulated dataset contained 5–12 layers with background velocity ranging from 2.00 to 4.00 km/s, with one salt body having a constant velocity value of 4.50 km/s and comprising 1600 models. The second dataset was simulated from the Society of Exploration Geophysicists (SEG) salt model with more complex backgrounds and velocity values that ranged from 1.50 to 4.48 km/s, and included 130 models. The test sets for the two cases were 100 and 10, respectively. In total, 29 shots were generated with 301 receivers. Through a comparative analysis of 1, 13, 21, 27, 29 shots, it was shown that the network achieved a more stable performance for a forward solution of 29 shots for each individual velocity model. After 100 epochs, an approximate PSNR of 23 dB and maximum SSIM of 0.38 was obtained with 29 shots. FCNVMB slightly outperforms FWI in some cases. In the reconstructed Earth models, FCNVMB properly positions the salt body but in many images wrongly predicts the geometrical boundaries.

A generative adversarial network (GAN) architecture, VelocityGAN, is proposed for VMB from seismograms [26] and comprises an encoder–decoder of five convolutional layers for the generator and a CNN architecture of five convolutional blocks, a global average pooling and fully connected layers for the discriminator. The loss function used for the discriminator is a Wesserstein loss with gradient penalty while for the generator is a combination of adversarial loss and image content loss based on mean absolute error (MAE) and mean squared error (MSE). Curved velocity models of varying layer angles and thicknesses were used, while velocity images also included faults. Testing was carried out on 50,000 acoustic models, with 3 shots per model and 32 receivers, with a split ratio of 7:2:1 for training:testing:validation. The model outperformed FWI, demonstrating that in order for the FWI to reach comparable performance, extra information from the seismic data is needed, requiring 32 simulation shots. The network was trained on models containing one fault and was tested on models with zero or two faults included. The obtained MAE was 75.85 for the 1-Fault CurvedData, but did not transfer well to the 0-Fault and 2-Fault CurvedData datasets with MAEs of 154.1138 and 242.33, respectively. SeismInvNet [31] is an encoder–decoder architecture for VMB which was originally evaluated on subsurface acoustic reconstruction through 12,000 (10,000 training, 1000 validation, 1000 testing) velocity models with 20 shots and 100 receivers, containing only curved layered structures and their corresponding synthetic seismic data pairs and achieved an SSIM of 0.95338 and MAE of 0.014962, compared to another encoder–decoder architecture named InversionNet [32] which achieved an SSIM of 0.824075 and MAE of 0.039713. An improved SeismInvNet was proposed [33] for a curved layer structure, fault and salt body models, resulting in the generation of 18,000 models (in a split ratio 10:1:1 for train:validation:test) for 20 seismic shots and 32 receivers. The obtained SSIMs for the curved layer structure, fault and salt body models were 0.905838, 0.844608, 0.893321, respectively, while the equivalent MAE results were 0.014703, 0.021852, 0.017017, respectively.

The PIX2PIX conditional GAN, with U-Net256 for the generator and PatchGAN for the discriminator, with the original architecture of [30], was demonstrated for DLI for marine seismic data [34], tested on two datasets. The first category of pairs comprised single-channel seismic images with the corresponding velocity images, while the second category included three-channel images synthesised by a post-stack seismic image, an average tomographic velocity and a two-way time grid, resulting in 2800 models split 20:8 for training:testing. The results indicated the superiority of the second set, quantified in terms of SSIM providing an average value of 0.994, and the generated velocity was visually iden-

tical to the ground truth, demonstrating that the geological structures were incorporated into the GAN velocity model. Visual analysis of the single set showed poor results.

In summary, a range of DL architectures have been demonstrated to work well for acoustic VMB with PSNR in the range 23–30 dB and SSIM in the range 0.8–0.99. However, these architectures all needed training on many thousands (1600–50,000) of synthetic data pairs, comprising salt bodies and heterogeneous models with faults, with the requirement of simulating 3 to 32 seismic shots per velocity model in order to achieve the aforementioned performance. This entails a significant computational cost for the procedure. Since previous studies have extensively compared DLI approaches with FWI and established that FWI required multiple sources in order to meet the performance of DLI, in this study, we will not repeat comparative FWI experiments for multiple shots but will focus on comparing the performance of proposed and benchmark DLI approaches for single-shot simulation, as well as include FWI performance for single shots in qualitative performance analyses, purely for the sake of completeness.

GAN-based architectures have recently become popular in subsurface reconstruction tasks because they enable the production of realistic and high-quality synthetic images through the competitive work of two networks, the generator and the discriminator [35]. For this reason, in order to evaluate performance with a minimum possible demand for synthetic simulations, we leverage upon the PIX2PIX architecture, with a Resnet-9 generator with performance evaluated for a range of heterogeneous multi-stratified velocity images created through variate techniques and for a limited number of training data (only 1000 training models) with a minimum number of associated seismic sources, i.e., single-shot simulation, and 301 receivers (one receiver for every 10 meters). We benchmark our results against the PIX2PIX image-to-image architecture, with the UNet-based generator and the UNet-based FCNVMB architecture. The U-Net architecture has previously demonstrated excellent performance in predicting solutions for spatiotemporal partial differential equations [36].

### 2.2. Data Generation for Training of Deep Learning-Based Inversion

Due to the limited availability of publicly accessed velocity models, most current seismic deep learning inversion studies either adopt simplistic assumptions for the synthetic velocity models or are based on heterogeneous models, occasionally with salt bodies but without presenting a specific methodology for construction. To the best of our knowledge, OpenFWI [37], Marmoussi [38], SEG BP models [39], SEAM (not freely publicly available) and SEG/EAGE salt and overthrust models are able to provide complex heterogeneous velocity models for testing and training. OpenFWI, an open-access dataset providing thousands of models, has been used for several DLI architectures, but without presenting the feature of salt bodies or taking into account the information from geological images or pixel intensities. Additionally, none of the aforementioned models have the flexibility to be modified to enable the generation of more custom velocity images to map any subsurface under investigation. Therefore, such public datasets cannot generate the thousands of distinct velocity models of one or more specific structures that are needed for training DLI for VMB models. To address the flexibility of generating custom datasets, there have been a number of recent attempts at generating training and testing sets of different types of structures for DLI models. These are summarised next.

A mathematical methodology for building compressional wave synthetic data such as dense-layer (curved)/fault/salt body velocity models, without much effort is proposed in [33], where mathematical steps for building curved velocity layers are provided as well as how to introduce faults and salt bodies. However, this paper does not present an exact methodology for simulating the influence of the upward movement of a salt body on the upper layers, instead setting random Gaussian curves for this purpose.

Another mathematical framework through which various complex velocity models with common geological structures such as folding layers, faults and salt bodies can be automatically generated is proposed in [40], where it is concluded (not demonstrated)

that the simulation functions could be replaced by actual geometrical data obtained from outcrops to simulate more realistic Earth models with stratifications. Images containing daily objects of no geological significance, such as animals, toys, vehicles, and houses from the COCO dataset [41] were used to generate velocity models in [42] for training the network. Whilst these natural images may be rich in structure and detail and hence produce detailed seismic data structures in forward modelling that may enhance the generalisation ability of the network during training, performance was average when tested on real geological models. In the presented cases, the main velocity body was formatted, but the total image was abstractedly predicted to miss a lot of structural information, resulting in poorer performance on such geological datasets. We suppose that this is because the natural images chosen were not representative of Earth images, which should better reflect geological subsurface information. Therefore, we propose to generate synthetic models based on true subsurface geological structures.

### 2.3. Summary of Contributions beyond State of the Art

The current data-driven VMB approaches are limited in that the results need additional information from the velocity models, as the networks require a number of simulated multiple shots per velocity model in order to achieve stable performance on variations in subsurface reconstruction images, further increasing the already many thousands of Earth models that are needed to train the networks. Additionally, most approaches have only been designed and validated on models that adopt simplistic assumptions for horizontal homogeneity or on complex 2D heterogeneous models without presenting a specific methodology for constructing the velocity array. Whilst there have been initial attempts to generate models from natural images, none have attempted to exploit images that present geometry from real geological layers combined with a mathematical methodology as one unit velocity building tool. In addition, no study has attempted to validate the data-driven VMB performance on velocity models with a large range of Earth structures in order to evaluate the stability of the prediction under a changing structural basis for the Earth model.

This paper attempts to address these gaps by first proposing a mathematical framework for generating simple 2D heterogeneous models (linear and curved/folded layers), models with salt bodies interacting with layers whose geometries are found on geological images and finally models that present high granularity, generated directly from geological images to be more representative of the real subsurface. Furthermore, we demonstrate the stability of performance across variations in stratifications for all five types of synthetic images using two types of state-of-the-art DL architectures for VMB, namely FCNVMB of [25] and the PIX2PIX conditional GAN network. Unlike in [34], where UNet was proposed for the generator and PatchGAN for the discriminator, we show that ResNet with nine residual blocks for the generator can lead to high performance predictions for the seismic inversion problem, against the most restrictive but cost-effective choice of simulating with one seismic source and using only 1000 models for the training set. Finally, we evaluate the generalisability of the data-driven VMB solution by simultaneously training and testing on models that were built on a different structural basis.

### 3. Methodology for Generating Synthetic Velocity Models and DLI architectures for VMB

In this section, a framework for the generation of synthetic 1D and 2D heterogeneous velocity models is presented, either relying solely on a mathematical procedure to model various geological structures or on processing a small amount of geological images. The aim is to provide representative synthetic data that can be used to train the DL solver for VMB. In this study, all simulations are carried out at a maximum depth of two kilometres, but it is also possible to use parts of the proposed framework for customised modelling in the near surface. We generate five synthetic datasets that differ in complexity of the generation procedures, in requirements of a geological image example, and in type of geological

structures that they model. In Section 4, we will compare the results on the unseen portion of real test dataset, obtained after training the networks with each of these five datasets.

Section 3.1 presents the mathematical formulation for the LINEAR 1D heterogeneous velocity model and the non-linear 2D heterogeneous FOLD velocity model, both generated without any image examples. Section 3.2 describes a 2D heterogeneous model, termed SALT, generated by mathematical manipulation of the geometries on a geological image, while Section 3.3 describes 2D heterogeneous models, termed IMAGE1 and IMAGE2, generated by mathematical manipulation of pixel intensities of the available geological images. When the velocity model generation is completed, the corresponding seismic shots are generated from the solution of the wave equation, as described in Section 3.4.1, and the training of the network takes place.

### 3.1. Proposed Methodology for Multilayered Structures with 1D and 2D Heterogeneous Velocity Models

First, we focus on generating two multilayered velocity models of varying thickness without any image samples. The first model is a simple linear model that presents horizontal homogeneity, while the second more complex model provides 2D heterogeneity simulating folded systems, as shown in Figure 2.

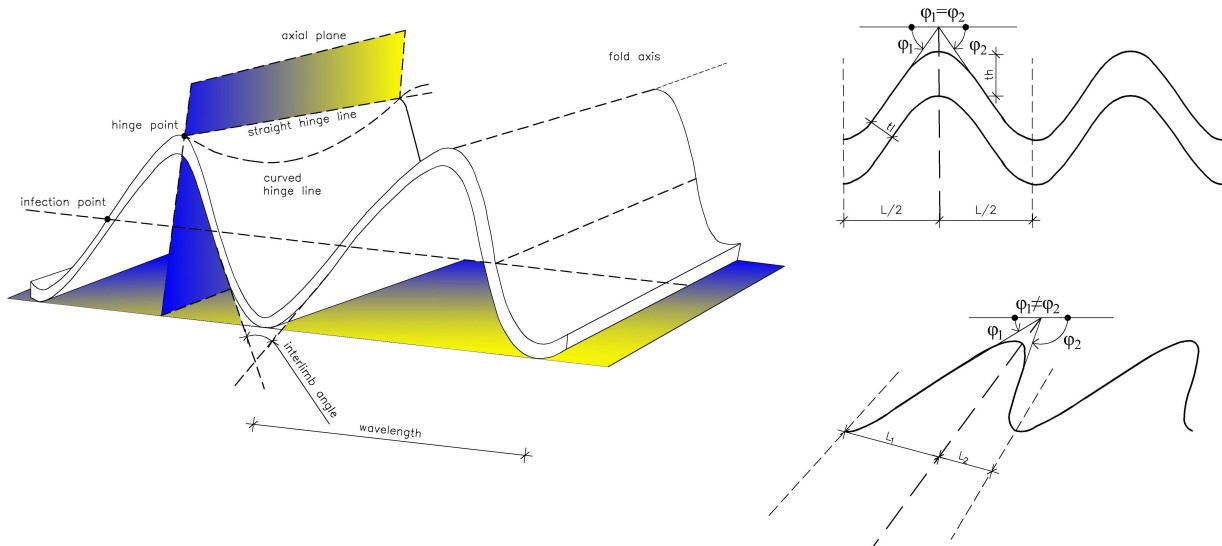

**Figure 2.** Characteristic properties of a folded structure. A fold can be characterised as cylindrical if it presents a straight hinge line or noncylindrical if not (**left**), and symmetrical if the two sides of the hinge are identical to each other (**upper right**) or nonsymmetrical (**lower right**).

Let $n_x$ and $n_z \in Z^+$ denote, respectively, the horizontal and vertical number of points of the elastic 2D grid plane. The proposed multilayer 1D heterogeneous subsurface image is generated as per Equation (1):

$$f_l(x) = a_l x + b_l, \tag{1}$$

where $x \leq n_x$, and $f_l(x) \leq n_z$, $l = 1, 2, \ldots$ indexes the ground layer, $x$ is the horizontal coordinate, $a_l$ is the inclination coefficient of the $l$-th layer, and $b_l$ is the constant value reflecting the thickness of the layer. By setting $a_l = 0$, we generate the first dataset, which is horizontally homogeneous with a number of horizontal layers. We will refer to this dataset as the LINEAR dataset.

Although very simple to generate, the LINEAR dataset does not capture characteristics of the folded structures.

Folded structures can be found in the subsurface as a stack of curved layers that present waviness. Under continuous processes, this curvature is changing through applied

deformation that creates natural folded systems. These structures are of great importance for geologists or petroleum engineers since they contain important information about Earth's tectonic processes or oil deposits and are connected to physical and chemical deformation procedures. Structural geologists have made a number of classifications of the folded structures according to the geometry of the fold, such as the characterisation of a symmetric fold when one side of the hinge is a mirror of the other side and the limbs have exactly the same length and anti-symmetric fold or cylindrical fold when the hinge line is a straight line parallel to the fold axis and noncylindrical fold (Figure 2). To introduce the characteristics of the folded structures such as waviness, next we use the sinusoidal function $s_l(x) = A\sin(wx)$, where $w = \frac{2\pi}{T}$, $T$ is the wave period and $A$ the amplitude. To simulate the asymmetry that folds may present, we generate a function composition ($\circ$) between $s_l(x)$ and $h_l(x) = wx + \sin(wx)/n$, where $n \neq 0$ is a real number that introduces asymmetry. Finally, in order to rotate this waviness according to the orientation of the initial linear velocity model, we add the function $f_l(x)$ leading to:

$$T_l(x) = (s_l \circ h_l)(x) + f_l(x), \tag{2}$$

where $x \leq n_x$, and $T_l(x) \leq n_z$, $l = 1, 2, \ldots$ indexes the ground layer, and $x$ is the horizontal coordinate. Equation (2) represents a non-linear shape function across horizontal coordinate $x$ for the layers $l$ that capture the waviness and asymmetry of the folded structures. Figure 3 shows examples of $T_l(x)$ for several values of $n$. The function is close to a sinusoidal function for $n = 10$. Unique velocity models (referred to as FOLD dataset) that present 2D non-linear heterogeneity, according to Equation (2), are generated by varying each of the five parameters of the model, namely, layer thicknesses ($b_l$), inclination angle ($a_l$), waviness amplitude ($A$), period ($T_l$), and asymmetry ($n$), one at the time. Namely, the velocity values are assigned between two successive layers according to the following equation:

$$V(i, j) = v_{l-1}, \ j = 1, \ldots, x_{max}, i = \lfloor (T_{l-1}(j)) \rfloor \ldots \lfloor (T_l(j)) \rfloor, \tag{3}$$

where $v_l$ is the velocity value of the $l$-th layer, picked from the interval [1.50, 4.00 km/s], making sure that the velocity increases with the depth; $l = 2, \ldots$ denotes the current layer, and $x_{max} \leq n_x$ is the domain of the shape function generated from Equation (2) along the horizontal $x$ axis.

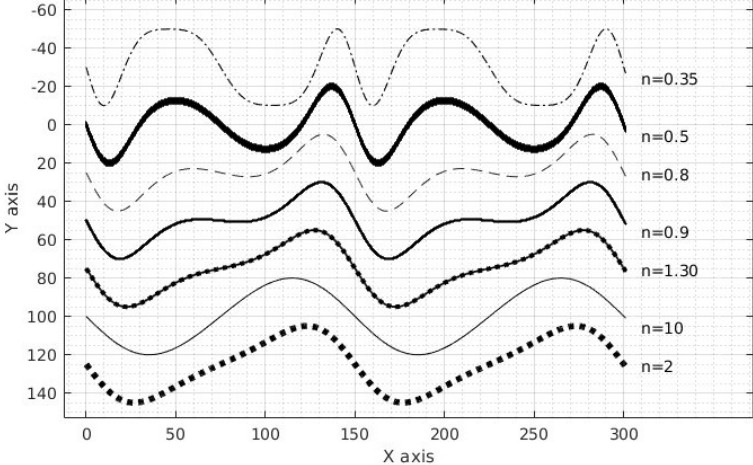

**Figure 3.** Example plots of function $T_l(x) = A\sin[wx + \sin(wx)/n] + a_l x + b_l$ for various values of $n$, where X axis shows values of $x$ and Y-axis shows values of corresponding $T_l$. For $n = 10$, the function is close to the sinusoidal function, while for other values, such as 1.30, the function is asymmetric.

### 3.2. Models Generated Using Pixel Positions and Salt Dome Upward Movement

Another geological structure of interest is the salt dome. Most geologists today believe that salt structures have resulted from plastic deformation of some existing soil layers or layers of salt of sedimentary origin. Salt structures exhibit upward movement, partly due to the difference in density between the salt layer and the overlying strata, although it is not clear whether the presence of some transverse compressing confinement stress is always required. Due to frictional forces, salt domes at the first stages of their development are most often circular in shape or are generally elliptical. Some of the salt masses are cylindrical, but downward enlargement is common, and most of the domes have the form of a truncated cone while the upper surface of the salt in most domes is flat or slightly convex upwards [43]. To improve the two velocity models described in the previous subsection towards capturing more complex geological structures, such as the salt dome, in this subsection, we propose a new type of model that is based on manipulating an existing geological image. Let $N$ and $M$ denote, respectively, the horizontal and vertical number of pixels in an image. For each image and each layer, $m$ pairs of pixel positions, as (row, column) or $(i_l, j_l)$, are selected randomly from the regions along the length of the curves/edges in the image. Each selected pixel location $(i, j)$ is scaled by the dimension of the 2D grid of interest, $(n_x \times n_z, n_x \leq N, n_z \leq M)$ as:

$$
\begin{pmatrix} x_1 & z_1 \\ . & . \\ . & . \\ . & . \\ x_m & z_m \end{pmatrix}_{m \times 2}
=
\begin{pmatrix} j_1 \frac{n_x}{N} & i_1 \frac{n_z}{M} \\ . & . \\ . & . \\ . & . \\ j_m \frac{n_x}{N} & i_m \frac{n_z}{M} \end{pmatrix}_{m \times 2}
\tag{4}
$$

This way, each of the $m$ pixel pairs $(i_l, j_l), i_l \leq M, j_l \leq N$ is mapped into a coordinate $(x_l, z_l), x_l \leq n_x, z_l \leq n_z$. Subsequently, the shape functions that describe the geometry of the layers are approximated according to the following equation:

$$
f_l(x) = \sum_{k=0}^{n} a_k x^k,
\tag{5}
$$

where $n$ is the degree of the polynomial function, and $l$ indexes the constructed layer. Coefficients $a_k$ are estimated using the $m$ selected pairs of image pixel positions through solving the following equation for $\mathbf{a}$:

$$
\mathbf{x}_{coord}^T \mathbf{x}_{coord} \mathbf{a} = \mathbf{x}_{coord}^T \mathbf{z}_{coord}
\tag{6}
$$

where:

$$
\mathbf{x}_{coord} = \begin{pmatrix} 1 & x_1 & . & x_1^n \\ . & . & . & . \\ . & . & . & . \\ . & . & . & . \\ 1 & x_m & . & x_m^n \end{pmatrix}_{m \times (n+1)}, \mathbf{z}_{coord} = \begin{pmatrix} z_1 \\ . \\ . \\ . \\ z_m \end{pmatrix}_m, \mathbf{a} = \begin{pmatrix} a_0 \\ . \\ . \\ . \\ a_n \end{pmatrix}_{n+1}
\tag{7}
$$

During the construction process, small smooth thickness variation of the background layers are preformed, as shown in Figure 4, so that all generated velocity models are unique. Velocity values belonging to the desired interval are assigned according to Equation (3) by substituting $T_l(x)$ with $f_l(x)$ obtained by Equation (5).

The procedure described next, through Equations (8)–(14), is used to optionally model the upward intrusion of salt structures in any velocity model.

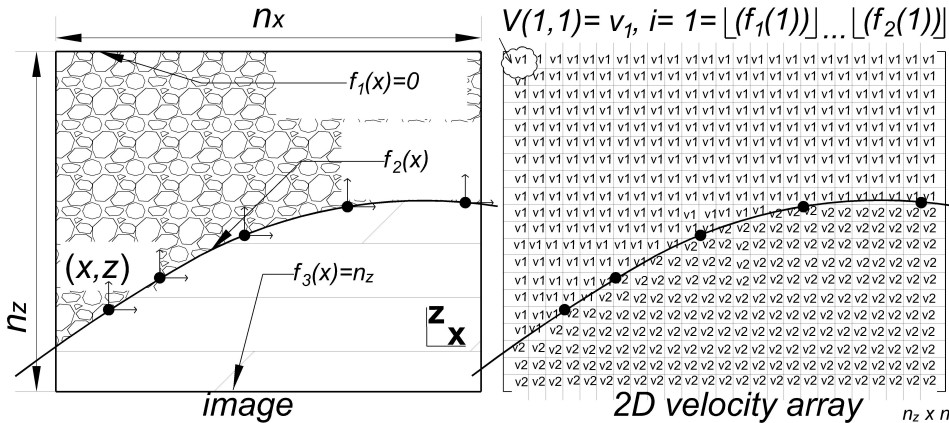

**Figure 4.** Shape functions from stratified geometries found on geological images. The vertical $z$ coordinate of the points is slightly varied in iterative steps, and shape functions (Equation (5)) are calculated repeatedly in order to introduce small geometric variations.

First, we describe mathematically how we simulate the interaction between the salt dome and the existing geological layers, due to its upward movement, with simple relations derived from differential geometry. The points of the induced salt dome are assumed to belong to the 1D Gaussian function as in prior work [33,40] in order to model the geological structures described above for a circular horizontal section and a vertical section of a truncated cone with a convex upward upper surface. To simulate the variation of thickness along the layer as the dome rises up, a reduction factor for every affected layer $l$ is defined as $p_l = \frac{t_h}{t_l}$ in the interval [0.25, 0.7], where $t_h$ represents the thickness at the point of the highest curvature, and $t_l$ is the thickness at the boundaries of the layer (as shown in Figure 2, upper right side). To simplify exposition, we assume that only two layers are affected by the upward intrusion of the salt dome. For the first layer affected mostly by the movement of the dome, $p_l$ is heuristically set to 0.67, while for the second layer, $p_l$ is set to 0.29. The salt dome's shape is then represented by:

$$q(x) = \frac{A}{\sigma\sqrt{2\pi}} e^{-0.5[(x-x_{ins})/\sigma]^2} \tag{8}$$

and for the interaction with the upper layers, the next equation is used separately for every affected layer, incorporating the reduction factor $p_l$ for the thickness variation calculated as:

$$n_l(x) = p_l q(x) \tag{9}$$

where:

- $A$ is the amplitude of the moving salt dome;
- $x_{ins}$ is the insertion point and centre of the dome;
- $\sigma$ is the dome shape parameter.

To approximate the impact of the vertical upward intrusion of the salt body on the layer, the vector $\mathbf{P} = (x, z)$ containing all the horizontal and vertical points of the affected layer is transformed to $\mathbf{P}' = (x', z')$ by the component addition of the displacement vector $\mathbf{T} = (\alpha, \gamma)$, that is, $x' = x + \alpha$, $z' = z + \gamma$, where $z = f_l(x)$ as defined in Equation (5) for $x \in [0, n_x]$, $n_x \in \mathbb{Z}^+$.

In order to obtain the components of the displacement vector, we transform the positions from the general structural system $x - z$ to the local system $t - n$, where $t$ is a

tangential and $n$ a normal axis to the layer's shape function at point $C$. For the triangle $ABC$ in Figure 5, which shows a vertical upward intrusion of a salt body, it can be observed that:

$$\widehat{ACB} = \widehat{tCx} \tag{10a}$$

$$\hat{\varphi} = atan(dz/dx) \tag{10b}$$

$$ds = (dx^2 + dz^2)^{1/2} = dx(1 + (dz/dx)^2)^{1/2}. \tag{10c}$$

By integrating both sides of Equation (10c) over the interval $[x_1, x_2]$ along the $x$ axis, as in Figure 5, we calculate the length on the layer along the tangential axis as follows:

$$s_{x_1 x_2} = \int_{x_1}^{x_2} (1 + (dz/dx)^2)^{1/2} dx. \tag{11}$$

By substituting $x = s_{x_1 x_2}$ into Equation (9), we calculate the distance (AC) (Figure 5):

$$(AC) = n_l(s_{x_1 x_2}) = \frac{p_l A}{\sigma \sqrt{2\pi}} e^{-0.5[(\int_{x_1}^{x_2}(1+(dz/dx)^2)^{1/2}dx - x_{ins}]/\sigma)^2}. \tag{12}$$

Furthermore, looking at the left side of Figure 5, BC and AB distances can be calculated as:

$$\gamma = (BC) = n_l(s_{x_1 x_2})cos(\hat{\varphi}) \tag{13a}$$

$$\alpha = (AB) = n_l(s_{x_1 x_2})sin(\hat{\varphi}). \tag{13b}$$

Finally, we obtain:

$$x' = x + n_l(s_{x_1 x_2})sin(\hat{\varphi}) \tag{14a}$$

$$z' = z + n_l(s_{x_1 x_2})cos(\hat{\varphi}). \tag{14b}$$

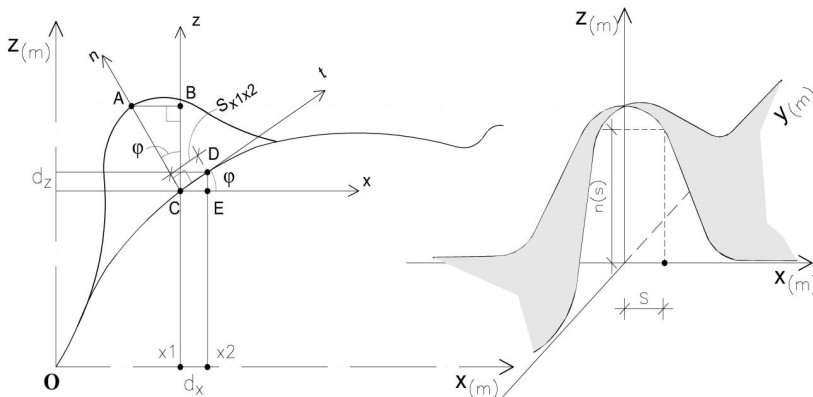

**Figure 5.** Vertical upward intrusion of a salt body. The axis $Ct$ is tangential to the layer's shape function, while $Cn$ axis is the normal at point $C$. The salt dome's shape is presented as an 1D Gaussian function which is a 2D vertical section of the imaginary 3D salt dome.

　　The dataset, which we refer to as SALT, is generated as per the above procedure to simulate the interaction of the upward movement with the two deepest layers whose shape function was generated according to the previous steps (Equations (5) and (6)) from the geological images. In parallel, small changes in the pixel positions were introduced, and the shape functions of the background layers were calculated iteratively according to Equation (5) so that thickness variation of the background layers can take place to generate a variety of image examples. At the same time, the training dataset was separated into three parts where the velocity range of the background layers was defined to belong to the three following intervals [1.5, 2.9], [1.8, 4.2], and [1.5, 4.0], while the salt dome whose amplitude and shape is varying maintained a constant velocity of 4.50 km/s in all

simulations. The results of this simulation process for the SALT dataset can be seen in Section 4.3.

### 3.3. Proposed Methodology for Direct Earth Model Generation from Pixel Intensities

Next, we describe the final proposed method to generate training velocity model samples using geological images of stratified content. The method ensures, on the one hand, that the velocities extracted from the pixel intensities present the maximum value at the greatest depth of the stratified image, and on the other hand, that the values fall within the desired interval denoted by $[V_L, V_U]$ km/s. The following steps are used to generate this dataset:

- Generate thickness variations in the geological images with elastic displacements picked from Gaussian distribution and applied to the four corners of the image and in the centre so that a variety of geometrical complexities is introduced (see Figure 6).
- Transform each three-channel RGB image into a single-channel greyscale image to make a 1:1 map between luminance and velocity value. Then, apply a Gaussian blur filter to achieve smoother velocity transition between the layers and to avoid sharp edges between the layers.
- Resize the images to $n_x \times n_z$, and let $p(i, j)$ be the image pixel, where $i = 0, \ldots, n_z$, $j = 0, \ldots, n_x$.
- To ensure that the bounds of the interval are used as velocity values as well, we calculate the maximum and the minimum pixel intensity, denoted as $p_{max}$ and $p_{min}$, respectively, and then transform pixel $p(i, j)$ to pixel $a(i, j)$ as follows:

$$a(i,j) = \begin{cases} 1 - \frac{p(i,j) - p_{min}}{255} & , p(i,j) < \frac{p_{max} - p_{min}}{2} \\ \frac{p_{max} - p(i,j)}{255} & , p(i,j) \geq \frac{p_{max} - p_{min}}{2} \end{cases}$$

- Define velocity array **V** with pixels $V(i, j)$ at the position $(i, j)$ according to:

$$V(i,j) = [1 - a(i,j)]V_L + a(i,j)V_U. \tag{15}$$

To ensure that the velocity model will have an increase in velocity with the chosen subsurface depth, whole images or parts of images that present stratified layers with colour grading are selected. When pixel intensities at the bottom layers of the grayscale image part are darker than the upper layers, Equation (15) is used, but when the bottom layers are lighter, we swap $V_L$ and $V_U$.

Two distinct datasets are generated using the aforementioned approach, termed IMAGE1 and IMAGE2, with geometrical variations after the application of elastic displacements according to Gaussian distribution and velocity variation with different interval bounds $V_L$ = 1.5, 1.8, 1.5 km/s and $V_U$ = 3.5, 3.8 and 4.3 km/s. IMAGE1 presents a less heterogeneous subsurface compared to IMAGE2 that models intensive heterogeneity, as shown in Figure 6, where the top row shows two example samples from IMAGE1 and the bottom row shows two example samples from IMAGE2. For the IMAGE1 examples the velocity bounds are $V_L$ = 1.5 and $V_U$ = 3.5, $V_L$ = 1.8 and $V_U$ = 3.8 km/s, while for the IMAGE2 examples, the velocity bounds are $V_L$ = 1.5 and $V_U$ = 3.5, $V_L$ = 1.5 and $V_U$ = 4.3 km/s.

### 3.4. PIX2PIX for Data-Driven VMB

In this section, we first describe how the five distinct seismic datasets (inputs) generated as per the proposed methodology are split into training and test sets for input into the DL architecture, as well as the velocity models (targets). Second, the PIX2PIX architecture hyperparameters for both the U-Net256 and ResNet-9 generator implementations, as well as the PatchGAN implementation for the discriminator and FCVMB DL solutions are presented. Finally, the performance metrics are described.

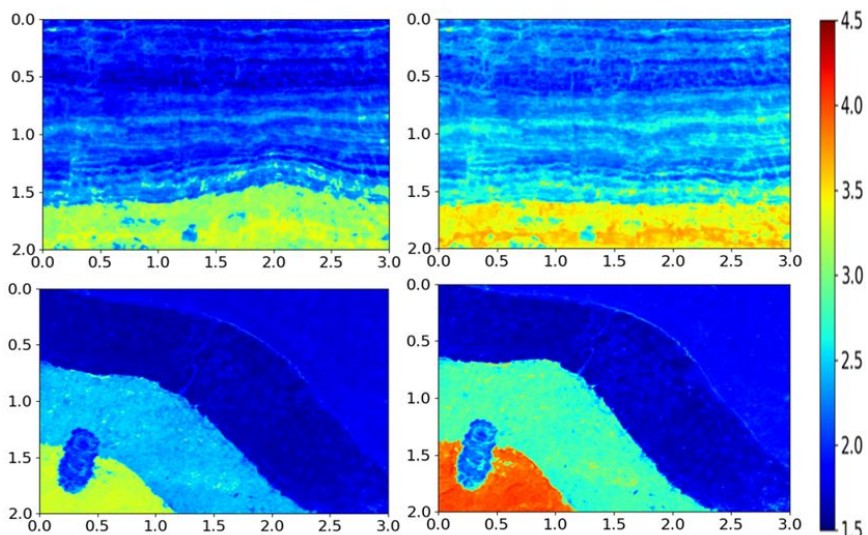

**Figure 6.** Models generated from pixel intensities of the used geological images. Geometrical variation is added through the application of elastic displacements according to Gaussian distribution, while velocity variation is added through the change in interval boundaries $V_L$, $V_U$ in Equation (15).

The original PIX2PIX architecture of [30] was demonstrated for the VMB problem with promising results on marine seismic data in [34]. For the seismic inversion problem, the function of a cGAN involves two deep networks: the generator and the discriminator. The generator under the control of the discriminator tries to learn a relationship in order to implement the prediction of the inverted velocity model and not to be rejected by the discriminator, which provides an extra benefit over the fully connected network architectures. Compared to a basic cGAN that creates images constrained according to a class vector, PIX2PIX is conditioned to the full content of one image without additive input noise [44]. The original network's generator uses the U-Net256 architecture, which adopts skip connections between the downsampling convolution branch and the upsampling deconvolution branch while the discriminator's architecture uses a patch with size $70 \times 70$ for the output, and all the responses from the convolutions are averaged. Although the U-Net256 implementation provides great results, in this study, we propose instead to use ResNet-9 for the generator. This is because ResNet-9 has previously demonstrated excellent performance in the computer vision field [45] and in cases of special medical datasets such as fundus images, where eye vessel networks were given as input, such as the seismic shots in the case of FWI [46].

### 3.4.1. Datasets

Five core training datasets were generated as follows: LINEAR as described in Section 3.1, FOLD as described in Section 3.1, SALT as described in Section 3.2, and IMAGE1 and IMAGE2 as both described in Section 3.3. In order to evaluate the generalisation of the DLI for VMB models for the above five datasets, we created three additional datasets. SYNTH1 comprised an equal proportion of the more popular velocity models seen in the literature of 333 samples from each of the LINEAR, FOLDED and SALT models for training. SYNTH2 comprised an equal proportion of 500 samples, each from the highly granular and more geologically complex IMAGE1 and IMAGE2 datasets for training. Finally, to evaluate the generalisability of DLI for VMB for all five proposed velocity models, the SYNTH3 dataset was created that comprised 200 samples each from all five proposed models. All the datasets created were split into the ratio 4:1:1 for the training, testing and validation samples, namely 1000 images for training, 250 images for validation and 250 images for testing. All pairs were created as a solution of the constant density acoustic wave equation in the time domain, as shown in Equation (16), for a Ricker source and receivers and grid points with physical parameters that can be seen in Table 1. We hypothesised that

the boundary conditions do not influence the performance of the DLI for VMB model, and we verify this by having two sizes for damping absorbing boundary layers of 100 (11 grid points) and 600 m (61 grid points), referred to as Boundary Conditions (BC) 1 and 2, respectively. The respective seismic shots are shown in Figure 7. Every simulation starts at time $t = 0$ to 2001 milliseconds. The velocity values increase with depth intervals [1.50, 4.50] km/s in steps of at least 0.20 km/s as we go deeper, up to 2 km.

$$S = \frac{\partial^2 u}{\partial t^2} - c^2 \nabla^2 u \qquad (16)$$

where:

- $u$, is the displacement field;
- $c$, is the P-wave velocity field;
- $S$, is the source function.

**Table 1.** Finite difference modelling in the time domain and the forward-solver's parameters.

| HOR.GRID | VER.GRID | BC1 | BC2 | Spacing | Sources | Frequency | Receivers | Simulation |
|---|---|---|---|---|---|---|---|---|
| 301 points | 201 points | 11 points | 61 points | 10 m | 1 | 10 Hz | 301 | 2 s |

All the corresponding seismic shots are generated through Devito [47], which is a domain-specific language for implementing high-performance, finite-difference, partial differential equation solvers.

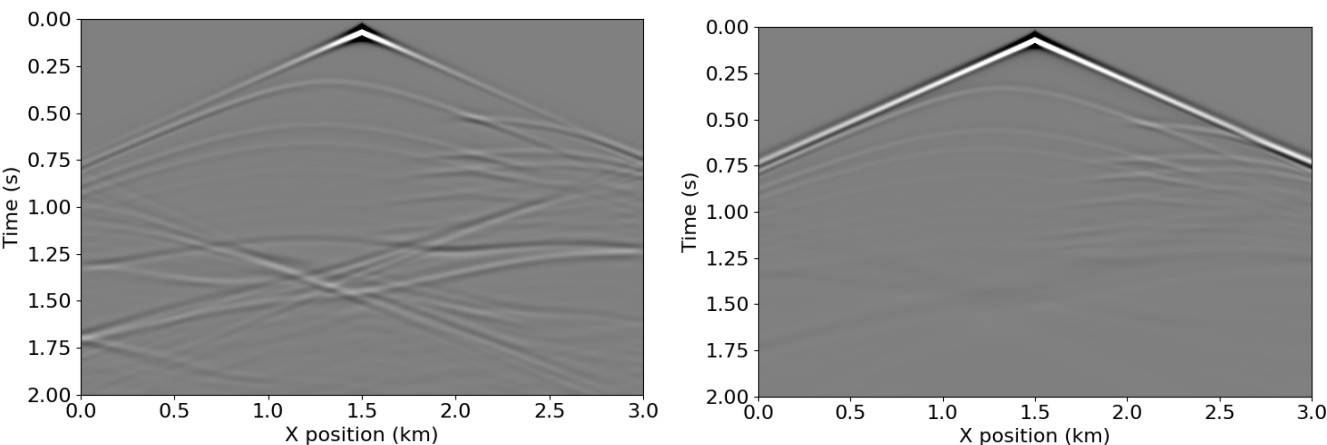

**Figure 7.** Seismic shots for BC1 with the size of the absorbing boundary layer equal to 11 grid points on the left and improved BC2 as per Table 1, with the absorbing boundary layer size equal to 61 grid points on the right.

Each set contains a feature that makes it stand out from the rest, while all datasets consist of three to eight layers with or without the addition of a salt structure. The LINEAR dataset contains linear stratification representing the case of 1D heterogeneity. To compensate for the simplistic presentation of the layers that remains horizontally homogeneous in this set, a stronger simultaneous variation in thicknesses was introduced in both the upper and lower layers compared to the other sets. In the LINEAR dataset, the models consist of velocity values in the range [1.50, 4.00 km/s]. The FOLD dataset introduces curvature to the layer boundaries, while the inclination, the thickness and the asymmetry also vary. The SALT dataset contains a salt structure of a constant velocity of 4.50 km/s, while the shape of the layers presented is derived from real images. The SALT models are simulated as follows: a thickness variation was applied as well as a variation-of-velocity value for the background layers, in parallel with application of the upward movement of salt bodies which have varying characteristics of amplitude and shape and calculation

of the interaction between them and the upper layers. The IMAGE1 dataset comprises velocities extracted from the fluctuating pixel intensities found in an image with mild 2D heterogeneity in the geological strata depicted, while the IMAGE2 dataset derives from an image with intense 2D heterogeneity. The velocity in IMAGE1 and IMAGE2 ranges in the interval [1.5, 4.3] km/s. The images that were used for velocity extraction were firstly elastically distorted with Gaussian distribution via the python AUGMENTOR, which is a library designed for image augmentation for machine learning purposes [48,49]. More specifically, for the images of the IMAGE1 dataset, a $3 \times 3$ grid with elastic movements of probability 1 and magnitude of 45 was applied, focused on the centre of the image and in all four corners. For IMAGE2, all the distortions were focused on the center of the image and have a magnitude of 90.

The models from the LINEAR dataset exhibit 1D heterogeneity and are the simplest compared to all the others examined in this study. In order to balance the lower level of structural complexity for this particular dataset, we incorporated strong variation in the thickness of the layers between the several velocity models to increase the variability and thus the difficulty of prediction. The models from the FOLD dataset present intense 2D heterogeneity induced by the use of the coefficient $n$ in $h_l(x)$ (see Equation (2)), which introduces asymmetry by inclination of the layers and by the waviness of the sinusoidal function, and they are clearly more complex than the LINEAR velocity models. The models from the SALT dataset also present lower geometrical complexity than the FOLD dataset, focused both on the interaction region due to the upward movement of the salt dome and on the simultaneous geometrical change of the background layers. The characteristic which further differentiates this dataset compared to the previous two is that small variability is also introduced in velocity ranges of the background layers, and thus, the velocities assigned to the background layers whose geometries also slightly vary do not remain constant across the 1000 training models. Velocity models belonging to datasets IMAGE1 and IMAGE2 also exhibit mild geometrical variation combined with variation in terms of velocity, also introduced between layers, similar to the SALT dataset. From a design point of view, the LINEAR and FOLD datasets introduce a stronger variety of change in geometry, while the remaining three datasets (SALT, IMAGE1 and IMAGE2) present a combination of milder geometric variety among the models and are combined with variability in the velocities which are assigned between the successive layers. In addition, the IMAGE1 and IMAGE2 datasets are differentiated from all the others because they are created through Equation (15), and thus, they present a granular look which appears because the velocity value changes from position to position within the same layer.

In addition to creating the Earth models of the eight datasets that were used for testing the prediction capability of the DLI architectures, blurred velocity models were also created with mathematical convolution of a Gaussian kernel, with the testing velocity arrays to be used as a starting model (can be seen in Section 4.3) for FWI, which adopted an adjoint-state-based gradient-descent method for the experiment, which was implemented through Devito in 25 iterative steps. For fair comparison with the DLI models, the simulation of the FWI was performed for a single source and lasted only a few seconds, while the dimensions of the grid and all the parameters of the physical problem were the same as those previously described in Table 1 for the forward modelling, to enable like-for-like benchmarking.

### 3.4.2. DLI for VMB Architecture Parameters

The five datasets were used for training and testing FCNVMB and PIX2PIX with the U-Net256 and ResNet-9 architectures. Seismic shots are input into FCNVMB with the velocity models as the target. Velocity models are arrays with dimension [201, 301], corresponding to a 2 km deep and 3 km long section with grid spacing of 10 m. Synthetic seismic shots are arrays with dimension [2001, 301], which represent 2001 ms over a distance of 3 km, as we are using surface receivers. Inputs to the PIX2PIX are two images, one for the velocity model and one for the seismic shot, each with dimension $256 \times 256$. Each image pair is combined into one image with size $512 \times 256$, while the direction for the training and the

testing phase is set from B to A, which means that the input of the seismic shot (B) to the target velocity model (A) is the same as in the FWI problem.

The values of the hyperparameters chosen for the DLI approaches in our experiments are the same as reported by [30] for both the U-Net256 and ResNet-9 generators of PIX2PIX and [25] for the benchmark FCNVMB, except for the learning rate, number of epochs, and the batch size, which are highly dataset-dependent and which are reported in Table 2. As is common practice, following heuristic evaluation using various hyperparameter values, we found that for the datasets examined in this manuscript, the reported hyperparameters in Table 2 are the ones that lead to the best performance (best predictions, less artifacts) of the DLI models. For PIX2PIX (U-Net256 and ResNet-9), the hyperparameter $\lambda$, which is a weighting regularisation factor in the L1 loss, was set to 100, and hyperparameter $\beta_1$ for the Adam optimiser was set to 0.5, as per default. Both architectures use the Adam optimiser, while FCNVMB uses an MSE loss function and PIX2PIX a conditional GAN loss with an $L_1$ term added that penalizes the distance between ground truth and synthetic outputs that match the input or not. After PIX2PIX initialisation, the PIX2PIX with ResNet-9 required 11,383,000 learnable parameters, while the original PIX2PIX with U-NET256 required 54,414,000. Thus, PIX2PIX with ResNet-9 is less complex, in terms of memory requirements, than PIX2PIX with U-Net256. For indicative purposes, for the stated training and test numbers of samples in Section 3.4.1 for the SALT dataset, for PIX2PIX with U-Net256, this is equivalent to 9.4 GB on a disk, whereas for PIX2PIX with ResNet-9, the memory requirement is only 2.3 GB. The equivalent memory requirement for FCNVMB is relatively lightweight at 621.3 MB. The equivalent indicative training and testing times for both PIX2PIX generators and FCNVMB are shown in Table 2. As observed, for the same number of images (250), the testing time for FWI is considerably higher than the DLI models but is on par with the total training and testing times of the DLI models. However, DLI models only need to be trained once for a particular subsurface structure, but they require a comprehensive dataset. The training and test times for ResNet-9 are slightly higher than those of U-Net256. Since the memory requirements for PIX2PIX with U-Net256 are significantly larger than those of PIX2PIX with ResNet-9, despite the slightly lower training and test times for the respective models, we conclude that PIX2PIX with U-Net256 is more complex overall than PIX2PIX with ResNet-9.

**Table 2.** Network hyperparameters and training time on a GPU NVIDIA RTX A4000.

| Network | L.RATE | Epochs | Batch Size | Training (min) | Testing (s) |
|---|---|---|---|---|---|
| PIX2PIX (U-Net256) | $2 \times 10^{-4}$ | 200 | 1 | 158' | 11.36 s |
| PIX2PIX (ResNet-9) | $2 \times 10^{-4}$ | 200 | 1 | 173' | 13.23 s |
| FCNVMB | $10^{-3}$ | 100 | 10 | 108' | 16.07 s |
| FWI | - | - | - | - | 12,500 s |

## 4. Experimental Results

In this section, we evaluate the performance of the proposed PIX2PIX with ResNet-9, benchmarked against the original PIX2PIX with U-Net256 and FCNVMB DL architectures, for each of the five proposed datasets (LINEAR, FOLD, SALT, IMAGE1 and IMAGE2) and the three SYNTH1, SYNTH2 and SYNTH3 datasets for the purpose of generalisability. In these cases, the DL architectures are trained on one dataset and are tested on an unseen portion of the same dataset. Evaluation is carried out through both quantitative and qualitative analyses. The quantitative analysis is based on the image quality metrics PSNR and SSIM, while the qualitative analysis is based on the characteristics of the predicted images and through the local SSIM maps which highlight the mispredicted zones.

### 4.1. Quantitative Evaluation Image Metrics: PSNR and SSIM

The metrics used for the quantitative comparison between the ground truth velocity images and the predicted images are PSNR and SSIM. PSNR is a quality image metric (expressed in dB); the larger its value, the better the quality of the predicted image being

considered, and it is defined as a relation between the maximum pixel intensity and the sum of squared differences of the pixel intensities of the predicted and the reference image:

$$PSNR = 20\log_{10}(\frac{MAX}{\sqrt{MSE}}) \tag{17}$$

where $MAX$ is the maximum pixel intensity value of the image, and $MSE$ is the mean squared error between the ground truth image $x$ and the predicted image $\hat{x}$ for the height $A$ (number of pixels) and width $B$ (number of pixels) of the compared images, calculated as:

$$MSE = \frac{\Sigma_{i=1}^{A}\Sigma_{j=1}^{B}[x(i,j) - \hat{x}(i,j)]^2}{AB}. \tag{18}$$

SSIM, on the other hand, is a quality metric that is a function of structural correlation $s$, luminance distortion $l$ and contrast distortion $c$ [50]:

$$SSIM(x,\hat{x}) = \frac{(2\mu_x\mu_{\hat{x}} + c_1)(2\sigma_{x\hat{x}} + c_2)}{(\mu_x^2 + \mu_{\hat{x}}^2 + c_1)(\sigma_x^2 + \sigma_{\hat{x}}^2 + c_2)}, \tag{19}$$

where $\mu$ and $\sigma$ represent the mean and the variance of each image, $\sigma_{x\hat{x}}$ is the covariance of $x$ and $\hat{x}$, and $c_1 = (0.01L)^2$, $c_2 = (0.03L)^2$ are two variables that stabilise the division with a zero denominator. $L$ is the dynamic range of the pixel intensities which equals $2^{bpp} - 1$, where $bpp$ are the bits used per pixel. SSIM is calculated for the entire image area. Additionally, a local SSIM value can be calculated for every pixel of the image and can be presented in a local SSIM map, such that the blurry regions of the predicted images are highlighted. The dark pixels in the local SSIM map indicate the blurry regions, while the bright ones represent regions that are less affected by blurring.

All metrics in the current study as well as the local SSIM maps were implemented via the following Matlab v. R2022b (9.13.0.2049777) functions: (1) ssimval for the SSIM value of the total image and ssimmap for the SSIM value per pixel as:

$$[ssimval, ssimmap] = ssim(predictedImage, referenceImage)$$

and (2) PSNR as:

$$peaksnr = psnr(predictedImage, referenceImage)$$

*4.2. Quantitative Results*

Tables 3 and 4 present mean PSNR values and mean SSIM values for each dataset for both PIX2PIX generator implementations (original U-Net256 and proposed ResNet-9) and FCNVMB for BC1 and BC2, respectively. Standard deviations are also calculated to indicate the stability of performance.

Table 3 shows that PIX2PIX in general has better PSNR performance than FCNVMB. Additionally, PIX2PIX shows stability in predicting the velocity model among various datasets which are based on a different structural methodology, as reflected in the significantly smaller velocity fluctuations in relation to the average value of PSNR. Figure 8 shows the change in PSNR values for PIX2PIX(ResNet-9) and FCNVMB for 250 testing models for IMAGE2 (on the left side of the diagram) and FOLD (on the right side of the diagram) datasets. The PSNR values for the FCNVMB network are in blue colour and show a strong fluctuation around the mean value, which is represented through a black line. The standard deviation for the particular dataset is expressed as the distance between the yellow line and the black line. For PIX2PIX, the PSNR values can be seen in green color and fluctuate less around the black line, which represents the mean value. The red lines represent the standard deviation of PSNRs for PIX2PIX.

**Table 3.** Results in terms of mean PSNR and SSIM with standard deviation values for PIX2PIX and FCNVMB averaged over 250 models for BC1 with 11 grid points of damping boundary layer. The 1st column contains the name of the examined dataset, the 2nd the adopted image metric (mean values), the 3rd and 4th column the results for PIX2PIX with U-Net256 and ResNet-9 for the generator, and the 5th column the results for FCNVMB. The best performance for each dataset is highlighted in bold.

| DATASET | METRIC | PIX2PIX (U-Net256) | PIX2PIX (ResNet-9) | FCNVMB |
|---|---|---|---|---|
| LINEAR | PSNR | 23.85 ± 3.51 | **24.72 ± 3.27** | 24.62 ± 2.49 |
| | SSIM | 0.99 ± 0.008 | **0.99 ± 0.005** | 0.98 ± 0.014 |
| FOLD | PSNR | 21.14 ± 1.89 | **23.15 ± 2.26** | 21.02 ± 4.42 |
| | SSIM | 0.97 ± 0.015 | **0.98 ± 0.012** | 0.95 ± 0.055 |
| SALT | PSNR | 23.64 ± 2.24 | **26.90 ± 1.92** | 14.85 ± 5.29 |
| | SSIM | 0.98 ± 0.009 | **0.99 ± 0.003** | 0.81 ± 0.146 |
| IMAGE1 | PSNR | **29.04 ± 1.71** | 27.11 ± 0.97 | 17.34 ± 1.40 |
| | SSIM | **0.99 ± 0.002** | **0.99 ± 0.002** | 0.94 ± 0.019 |
| IMAGE2 | PSNR | **24.55 ± 1.27** | 22.59 ± 1.27 | 15.76 ± 2.76 |
| | SSIM | **0.98 ± 0.006** | 0.97 ± 0.011 | 0.91 ± 0.050 |
| SYNTH1 | PSNR | 22.46 ± 2.59 | **23.84 ± 2.26** | 19.61 ± 2.86 |
| | SSIM | 0.97 ± 0.013 | **0.98 ± 0.008** | 0.95 ± 0.036 |
| SYNTH2 | PSNR | **25.36 ± 2.01** | 22.35 ± 1.13 | 15.25 ± 2.26 |
| | SSIM | **0.98 ± 0.007** | **0.98 ± 0.008** | 0.90 ± 0.053 |
| SYNTH3 | PSNR | **23.26 ± 2.42** | 22.59 ± 2.48 | 13.99 ± 2.51 |
| | SSIM | **0.98 ± 0.011** | **0.98 ± 0.011** | 0.84 ± 0.074 |

**Table 4.** Results in terms of mean PSNR and SSIM with standard deviation values for PIX2PIX and FCNVMB averaged over 250 models for BC2 with 61 grid points of damping boundary layer. The 1st column contains the name of the examined dataset, the 2nd the adopted image metric (mean values), the 3rd and 4th column the results for PIX2PIX with U-Net256 and ResNet-9 for the generator, and the 5th column the results for FCNVMB. The best performance for each dataset is highlighted in bold.

| DATASET | METRIC | PIX2PIX (U-Net256) | PIX2PIX (ResNet-9) | FCNVMB |
|---|---|---|---|---|
| LINEAR | PSNR | **28.19 ± 4.99** | 23.08 ± 1.85 | 27.47 ± 2.99 |
| | SSIM | **0.99 ± 0.003** | **0.99 ± 0.004** | 0.99 ± 0.006 |
| FOLD | PSNR | 20.77 ± 1.53 | **22.48 ± 2.16** | 21.08 ± 4.20 |
| | SSIM | 0.96 ± 0.012 | **0.97 ± 0.012** | 0.95 ± 0.063 |
| SALT | PSNR | 23.59 ± 3.44 | **25.78 ± 1.75** | 13.79 ± 5.04 |
| | SSIM | 0.97 ± 0.027 | **0.99 ± 0.004** | 0.77 ± 0.144 |
| IMAGE1 | PSNR | **27.84 ± 2.75** | 24.14 ± 0.87 | 16.69 ± 4.00 |
| | SSIM | 0.96 ± 0.012 | **0.99 ± 0.003** | 0.91 ± 0.052 |
| IMAGE2 | PSNR | **21.98 ± 0.80** | 21.45 ± 1.38 | 18.31 ± 2.16 |
| | SSIM | **0.97 ± 0.004** | **0.97 ± 0.010** | 0.94 ± 0.030 |

We hypothesised that DLI based on image pairs is not affected by boundary conditions. Comparing Table 4 with Table 3, it can be observed that the DLI models are not sensitive to changing the width of the absorbing boundary condition for all datasets, with only small variations in PSNR and SSIM for all datasets. We note that FCNVMB, which is based on learning non-linearity through mapping the values of the velocity models to values in the seismic shot (not image pairs such as PIX2PIX). slightly improves its performance for LINEAR and IMAGE2 datasets due to the reduced reflections, while the image-to-image PIX2PIX-based DLI models are mostly unaffected except for an increase in PSNR performance for the LINEAR dataset. The results overall do not differ significantly, and hence, we can conclude that the proposed DLI based on PIX2PIX with the ResNet-9 generator is not very sensitive to the effects of reflections from the boundaries.

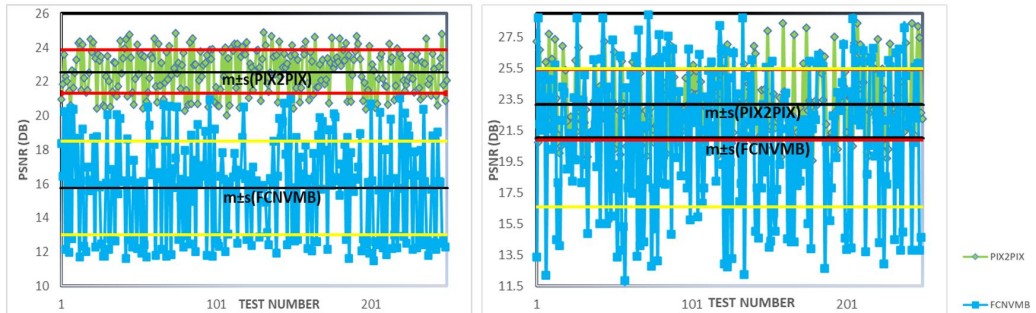

**Figure 8.** PSNR series (green for PIX2PIX (ResNet-9), blue for FCNVMB), meanPSNR (black lines) and meanPSNR $\pm$ *std* plots (yellow line for FCNVMB, red line for PIX2PIX) based on 250 predictions by PIX2PIX and FCNVMB. Results for IMAGE2 dataset are shown on the left. PIX2PIX (ResNet-9) exhibits a more consistent behaviour across various models with completely different characteristics. FOLD dataset results are shown on the right where the intensive instability of FCNVMB dominates the chart.

There is a noticeable drop in PSNR performance for FCNVMB as the velocity models increase in complexity—FCNVMB performs on par with PIX2PIX for the LINEAR models only, but its performance drops considerably for the other four more complex velocity models. On the other hand, PIX2PIX exhibits a more consistent behaviour which does not degrade across the datasets.

The SALT, IMAGE1 and IMAGE2 velocity models differ from LINEAR and FOLD models, as in addition to geometric variability, they also present variability in the velocity values assigned between the layers across the various velocity models on the dataset. In addition, IMAGE1 and IMAGE2 differ from the other three because they do not maintain a constant velocity across every layer of the velocity model. Thus, when intense geometrical variation takes place, PIX2PIX (LINEAR, FOLD) achieves similar or better performance compared to FCNVMB. When less intense geometrical variability is induced but velocity also varies, then PIX2PIX provides much better performance than FCNVMB. As we can observe in Tables 3 and 4, PIX2PIX can better handle the velocity variation than the geometrical variation, while FCNVMB does the opposite, and in all cases, PIX2PIX presents the best performance. Except for the most simplistic LINEAR model where all DLI for VMB models have similar SSIM, PIX2PIX also outperforms FCNVMB in terms of SSIM.

Quantitative metrics from Tables 3 and 4 clearly indicate that the PIX2PIX architectures consistently outperform the FCNVMB architecture for the DLI for VMB problem, under the same conditions. The PIX2PIX with ResNet-9 as the generator outperforms the PIX2PIX with U-Net256 as the generator for the FOLD and SALT datasets in terms of PSNR. This is also observed in the SYNTH1 dataset, which comprises models from these three datasets. On the other hand, the PIX2PIX with U-Net256 outperforms the PIX2PIX with ResNet-9 for the IMAGE1 and IMAGE2 datasets, as also observed in the SYNTH2 dataset. This relative PSNR performance is expected because U-Net256 has a more complex learning architecture than ResNet-9 and can hence better learn and distinguish the high granularity features of the IMAGE datasets. SSIM performance is consistently the best for all datasets with the proposed PIX2PIX architecture and the ResNet-9 generator. In general, ResNet networks tend to provide better colour rendering and more vibrant colours, as seen by the slightly better performance for the LINEAR, FOLD, SALT and SYNTH1 datasets. U-Net tends to be able to retrieve lost pixel information and provides more realistic images with higher pixel variation, as observed by the relatively better performance over ResNet-9 for datasets containing the IMAGE1 and IMAGE2 structures. However, when trading off complexity and performance, as demonstrated by SYNTH3 where the PSNR performance is similar for both U-Net256 and ResNet-9, we conclude that the PIX2PIX with ResNet-9 is the preferred approach for data-driven VMB. both PIX2PIX architectures have similar stability of performance as observed by the standard deviation of both the PSNR and SSIM metrics.

In addition to the image quality metrics, the training loss curves over time for the ResNet-9 and U-Net256 models are presented to further quantify the performance and to demonstrate successful learning. We show the discriminator loss on fake and real images as well as the generator loss. As can be observed in Figure 9, the generator loss $G_{L1}$ (red curve), which is related to the quality of the produced image itself, converges to a stable average. The discriminator losses also reach convergence, as seen from the $D_{real}$(blue) and $D_{fake}$(green) curves. The simplest LINEAR dataset is stable from the start. The MSE training loss curves for the FCNVMB benchmark, shown in Figure 10 for the IMAGE1 and IMAGE2 datasets, take longer to converge compared to the other simpler datasets. The training loss curves are in line with those of [25]. Finally, Figure 11 shows the convergence curves for LINEAR, FOLD, SALT, IMAGE1 and IMAGE2 for the FWI method. As expected, the method does not converge for the single shot except for the case of IMAGE2.

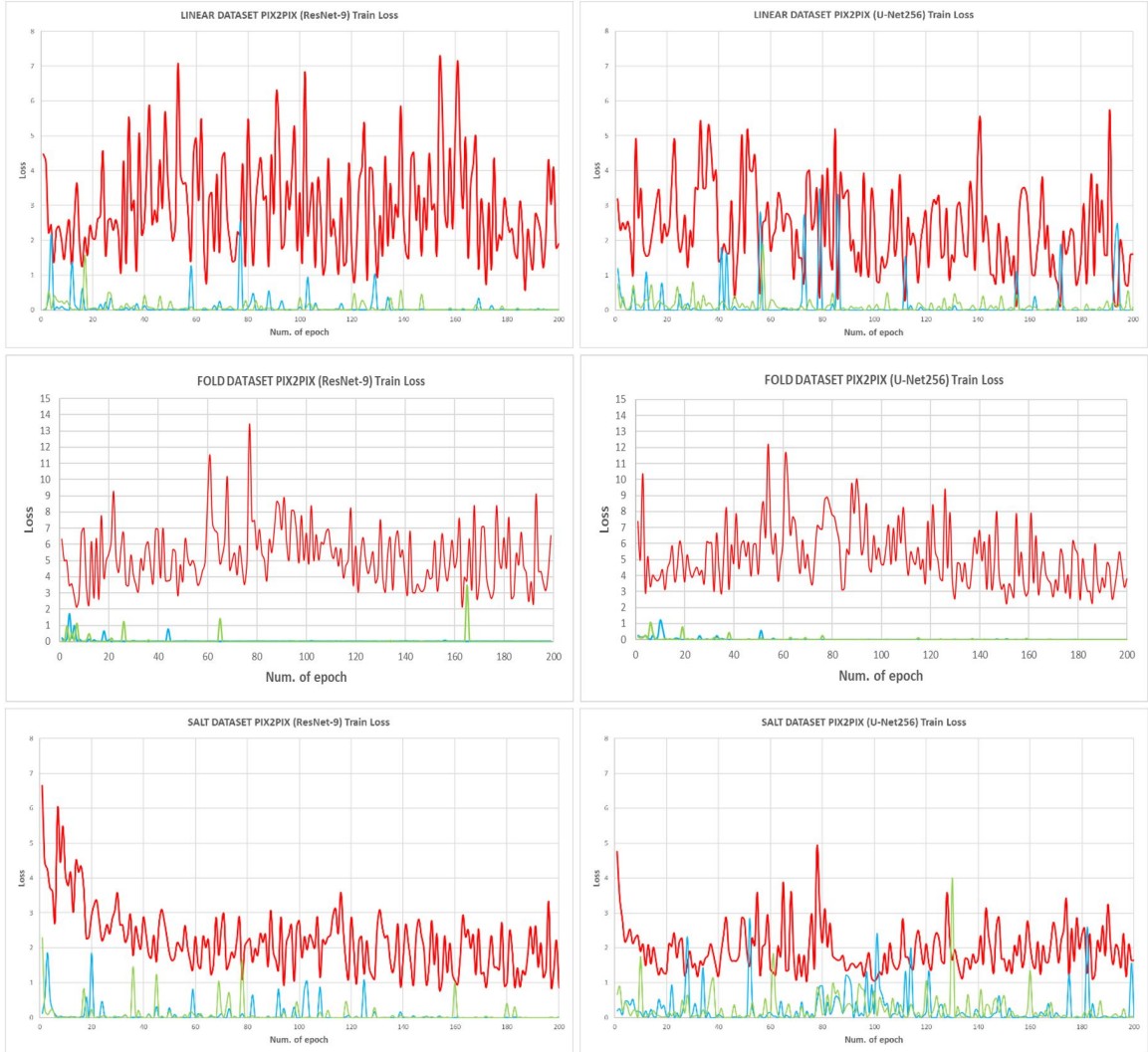

**Figure 9.** *Cont.*

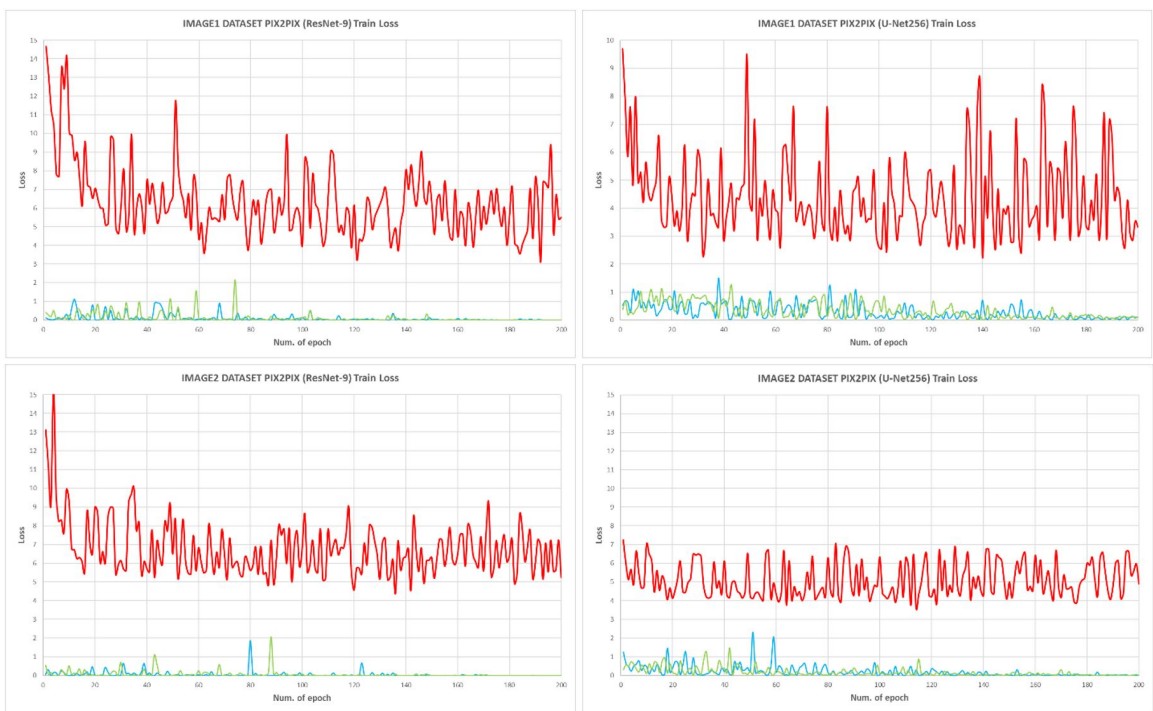

**Figure 9.** Train loss over time for PIX2PIX DLI architectures with ResNet-9 and UNet-256 generators, on the left and right, respectively, for the datasets, from top row to bottom: LINEAR, FOLD, SALT, IMAGE1 and IMAGE2. The red line shows the generator loss, while the blue and green lines represent the discriminator loss for real and fake images, respectively.

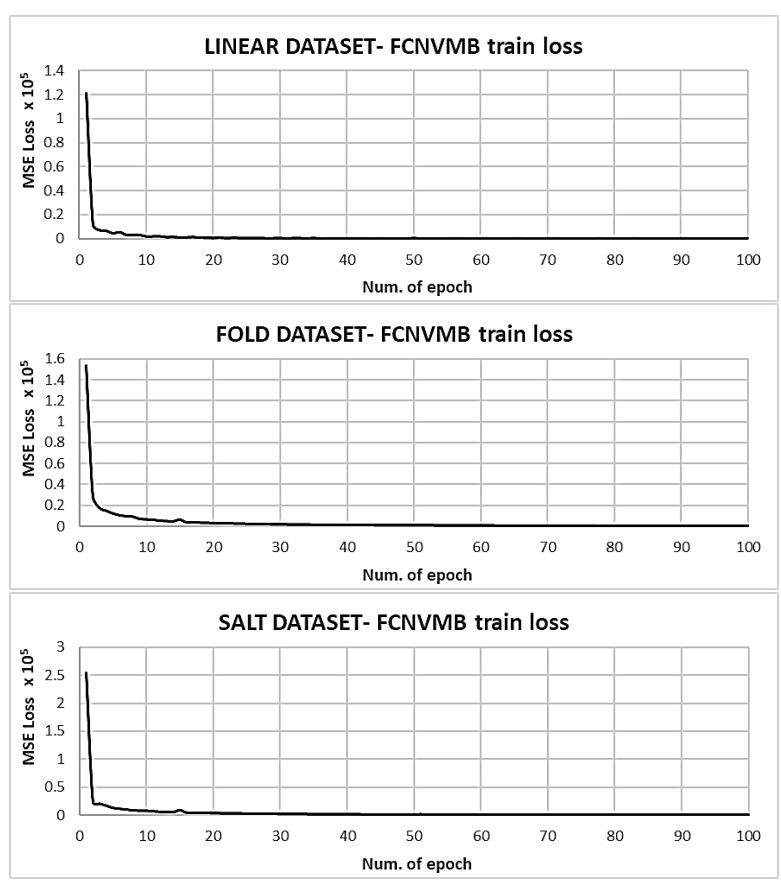

**Figure 10.** *Cont*.

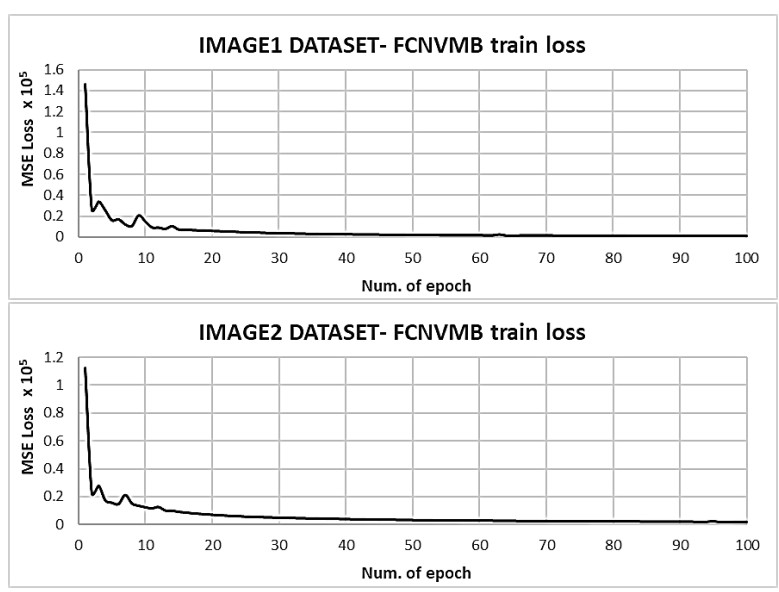

**Figure 10.** MSE training loss curves for FCNVMB model for the LINEAR, FOLD, SALT, IMAGE1 and IMAGE2 datasets.

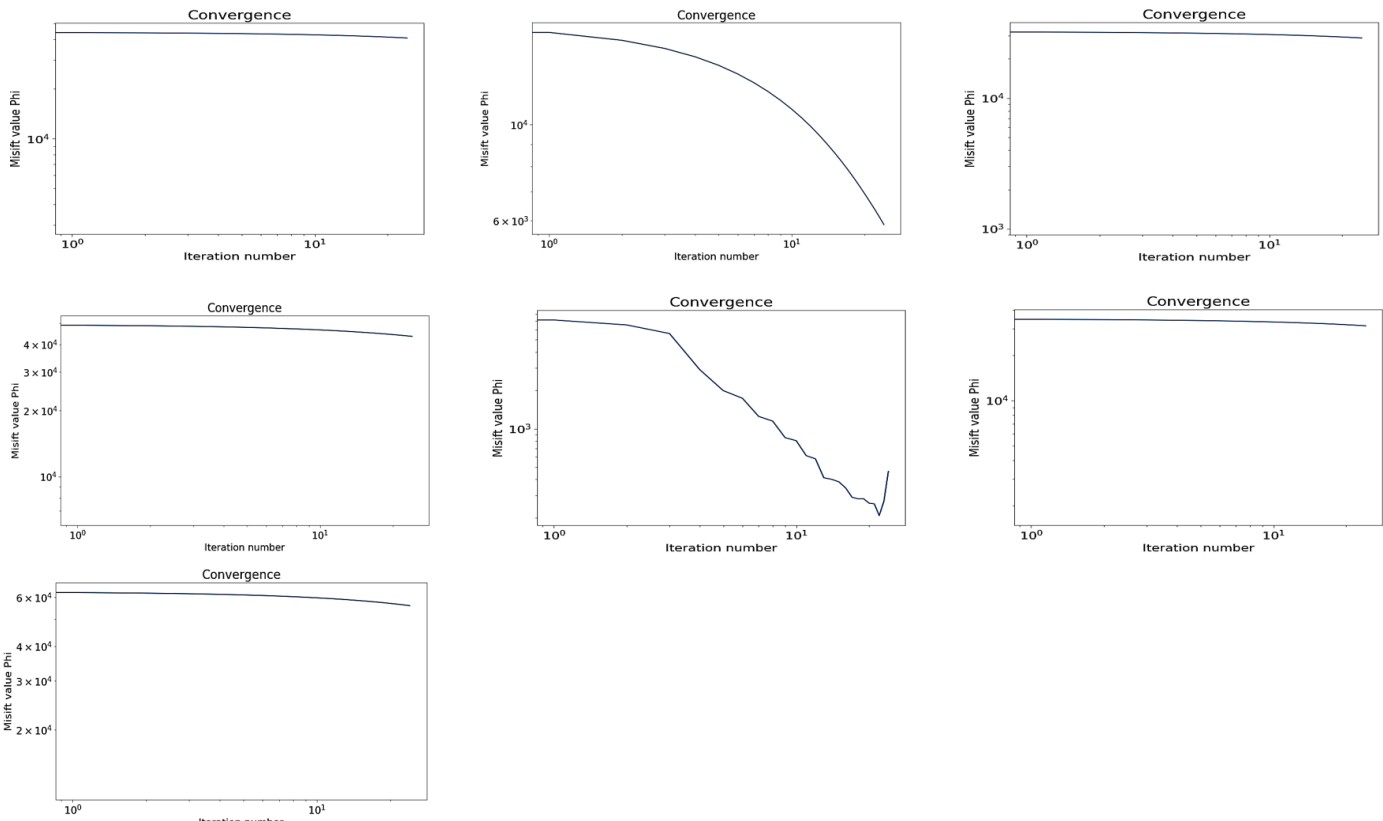

**Figure 11.** FWI convergence curves for all corresponding FWI models of the last column of Figure 12. From left to right, respectively, and from top row to bottom: FOLD, IMAGE1, FOLD, SALT, IMAGE2, FOLD and LINEAR (as seen on last column in Figure 12 from first to last row). The vertical axis of the plots expresses the misfit value of the function Phi subjected to minimisation through iterative cycles. All plots were obtained by Devito.

### 4.3. Qualitative Results

The qualitative performance analysis for DLI architectures was performed via visual observation of predicted velocity model images in relation to the corresponding ground truth. Additionally, a local SSIM map illustrates all blurry points of the velocity model that were predicted, and points that differed compared to the ground truth model were shown as dark pixels. Thus, by looking at the dark pixels, the human eye can more easily focus and identify the weaknesses of each prediction. Figure 12 shows the ground truth, predicted PIX2PIX (ResNet-9) BC1 experiment and predicted FCNVMB velocity model images, in the first, second and third columns, respectively. The fourth and fifth columns show the local SSIM maps for the predicted PIX2PIX(ResNet-9) BC1 experiment and FCNVMB velocity model images, respectively. The sixth column shows the starting model for FWI, and the seventh column shows the inverse velocity model image calculated from FWI.

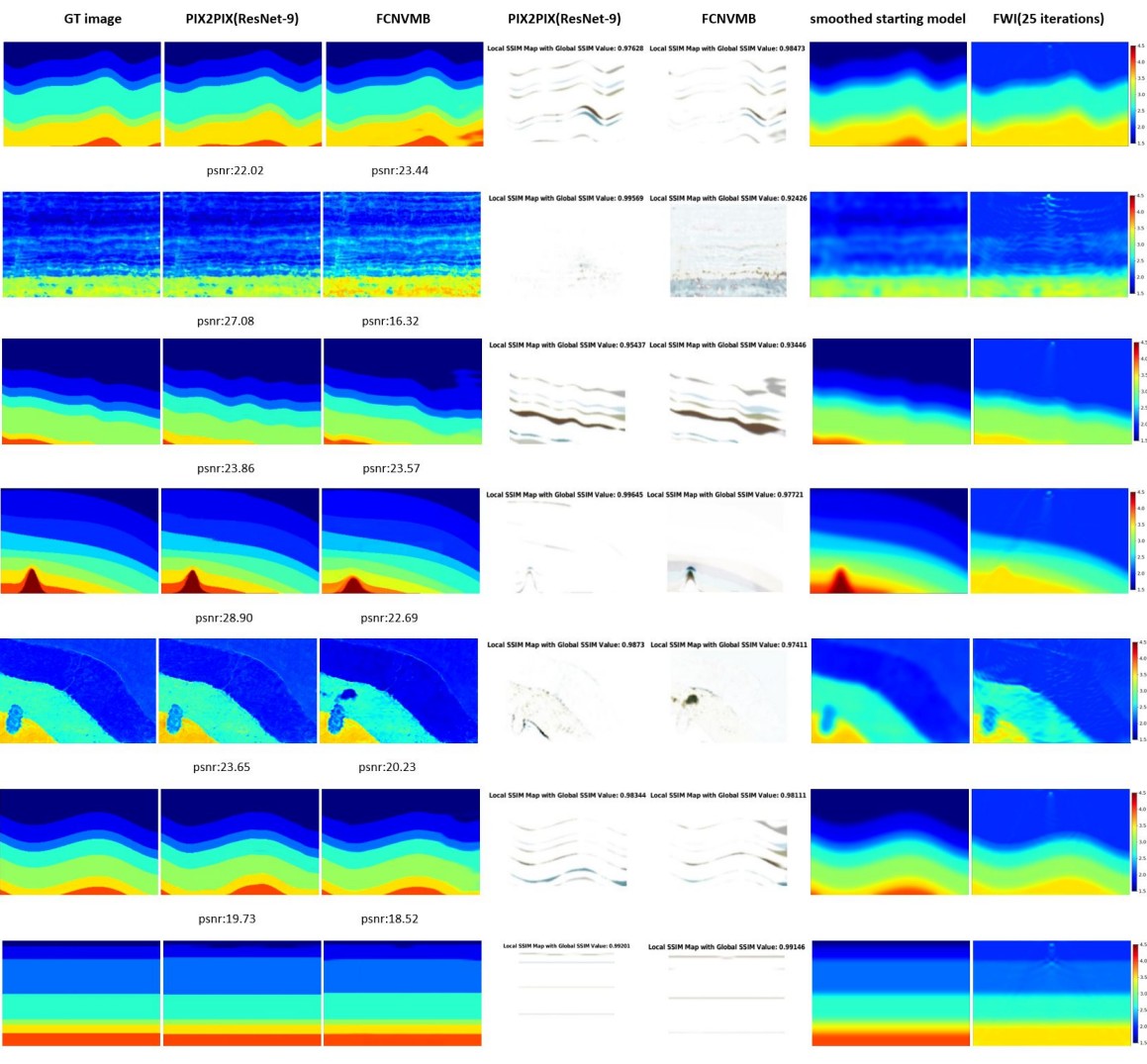

**Figure 12.** Predicted Earth models from PIX2PIX (ResNet-9), FCNVMB and FWI method (2nd column, 3rd and 7th row, respectively). FOLD (1st, 3rd, 6th rows), IMAGE1 (2nd row), SALT (4th row), LINEAR (7th row) and IMAGE2 (5th row) datasets can be seen in this Figure. Smoothed starting models (6th column) of the ground truth (GT) images (1st column) were used for full waveform inversion iterations (7th column). Local SSIM maps for PIX2PIX(ResNet-9) and FCNVMB can be seen in the 4th and 5th columns. The horizontal and vertical axes of the velocity models shown in this Figure are both in km. All velocity models are 3 km across and 2 km deep, while velocity values in all plots are in km/s.

For the least complex LINEAR dataset (7th row), compared to the ground truth, for most of the test models, both PIX2PIX and FCNVMB slightly mispredicted some of the layer thicknesses but generally made excellent predictions. As can be seen in the 7th row, 4th column, the small misprediction regarding the thickness of some layers by PIX2PIX is identified as a black horizontal line in the local SSIM map. The same can be seen on the 5th column for FCNVMB. FWI led to poor inversion results and convergence for the single-shot simulation, and as we can observe among most of the cases on the 7th column, it generally failed to accurately capture several layers and, most of the time, the deepest ones.

For the FOLD dataset, PIX2PIX mispredicted the deepest layer but generally provided a good result, while FCNVMB failed to accurately predict parts of the shape of the bottom layers, as observed in Figure 12 (1st, 3rd, and 6th rows). The local SSIM maps show the differentiation in the predicted layers' thicknesses for both networks. In addition, for the prediction of FCNVMB, where a blurred zone occurred at the right bottom side of the model (1st row) and at the middle (3rd row), the local SSIM maps highlighted the position with grey pixels for both cases. The FWI failed to identify the two upper zones, which are predicted as one unit zone, but also the deepest bottom layer.

For the SALT dataset, as shown in Figure 12 (4th row), PIX2PIX more accurately predicted the shape of the dome and the interacting zone (2nd column), as we can also observe through the lighter pixels in the local SSIM map (4th column). On the other hand, FCNVMB did not accurately predict the dome's shape or the interacting zone, as observed in the 3rd and 5th columns. This is why the local SSIM map for the FCNVMB captures the difference in the height of the dome. The FWI again failed to predict the deepest zone and to differentiate the upper two layers.

For IMAGE1, as observed in Figure 12 (2nd row), unlike PIX2PIX, FCNVMB missed the velocity range (3rd column), as can be also seen in local SSIM map as darker pixels across the entire surface of the Local SSIM map (5th column). The FWI performed very accurately for this dataset, providing comparable prediction to the DLI methods and capturing the velocity interval more accurately than FCNVMB. For IMAGE2, unlike PIX2PIX, FCNVMB made an incorrect velocity prediction for a small area centred at mid-depth of the model, Figure 12 (5th row, 3rd column), which can also be seen in the Local SSIM map as darker pixels (5th row, 5th column). The FWI performed again very accurately for the IMAGE2 dataset, providing comparable qualitative results to the DLI-methods.

Overall, PIX2PIX consistently outperformed FCNVMB and FWI, as observed by the well-defined (non-blurry with no artifacts or shape distortion) layer predictions in the velocity model and less dark pixelations on the local SSIM maps in the 4th and 5th columns. This observation aligns well with the observations made previously with the quantitative PSNR and SSIM metrics. In Figure 13, we attempted to compare the relative performance of the three DLI architectures with BC1 experiments for the SYNTH3 dataset by visual observation of the predicted models vs. the ground truth and the local SSIM maps. During quantitative performance analysis, we observed that PIX2PIX with ResNet-9 outperformed PIX2PIX with U-Net256 for the LINEAR, FOLD and SALT datasets. This can be explained by the distinct artifacts observed in the 2nd column for the first three rows of Figure 13, compared to the 3rd column. Similarly, as observed in the local SSIM maps, there are more dark contours in the 4th column compared to the 5th column for ResNet-9. On the other hand, during quantitative analysis, we observed that PIX2PIX with U-Net256 slightly outperformed PIX2PIX with ResNet-9 for the IMAGE1 and IMAGE2 datasets. As observed in Figure 13 in the 3rd and 4th rows, there is almost no visual difference between the predicted models in the 2nd and 3rd columns compared to the ground truth. This is also observed in the equivalent local SSIM maps. This reinforces our earlier conclusion that the proposed PIX2PIX with ResNet-9 is the best approach for DLI for VMB for the full range of velocity models when taking into account quantitative and qualitative performance evaluation, with the additional advantage that ResNet-9 is less complex than the U-Net256 architecture. Finally, as seen in the 4th column of Figure 13, FCNVMB does not achieve a comparable performance, since the predictions cannot accurately capture either the subsurface geometry

or the velocity range of the ground truth images, depending on the case. At the same time, the more heavily shaded areas that can be seen in the last column of Figure 13 compared to the 5th column of Figure 12 for the individual datasets (LINEAR, FOLD, SALT, IMAGE1 and IMAGE2) highlight the significant drop in performance regarding the predictions of the network for the SYNTH3 dataset.

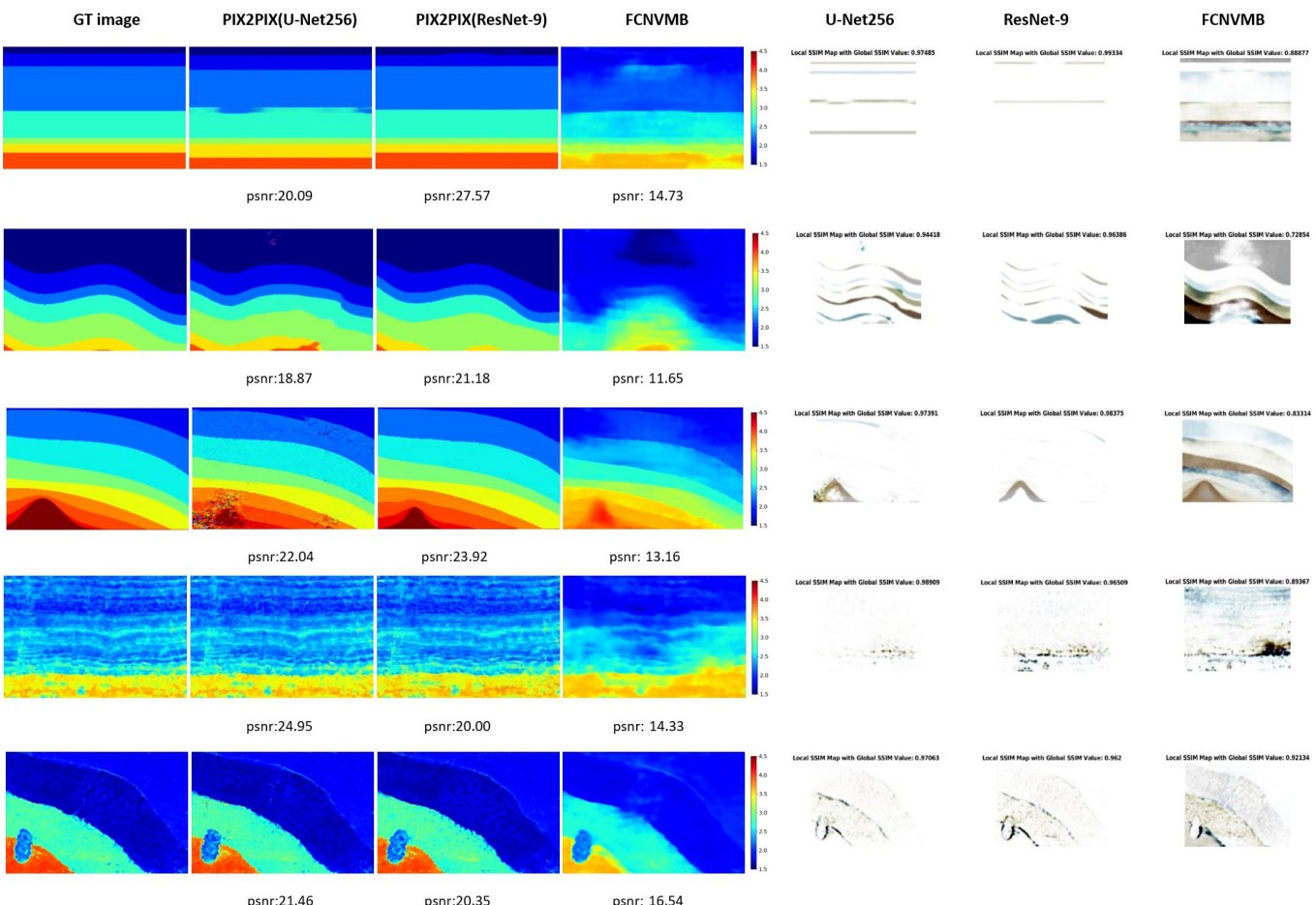

**Figure 13.** Predicted velocity models from SYNTH3 dataset for each of the five proposed datasets, showing an example for LINEAR (1st row), FOLD (2nd row), SALT (3rd row), IMAGE1 (4th row) and IMAGE2 (5th row). The 1st column shows the ground truth image, 2nd and 3rd columns show PIX2PIX with U-Net256 and ResNet-9, 4th column shows FCNVMB, and the 5th and 6th and 7th columns show the local SSIM maps for PIX2PIX with U-Net256, ResNet-9 and FCNVMB, respectively, against the ground truth (GT). The horizontal and vertical axes of the velocity models shown in this Figure are in km. All velocity models are 3 km long and 2 km deep, while the velocity values in all plots are in km/s.

## 5. Conclusions

In this paper, we proposed a complete methodology for synthetic generation of velocity models to enable the rigorous evaluation of deep-learning-based inversion for velocity model building via the custom generation of velocity models with various layers of complexity, allowing for folds, curvatures and salt bodies with multi-layer stratifications as well as more granular velocities with finer multi-layer stratifications, derived from geological images. The construction methodology that was proposed provides a variety of tools for simulating models, as it allows for the use of mathematical methodology with the parallel utilisation of geological images. We also adopted the PIX2PIX conditional generative adversarial network and proposed the less complex ResNet-9 architecture as a generator instead of the original U-Net256 architecture. The aim was to obtain competitive results

for seismic inversion, under a limited number of associated sources for a small training dataset. Rigorous quantitative and qualitative evaluations though PSNR and SSIM metrics as well as local SSIM maps highlight the ability of PIX2PIX with ResNet-9 to lead to the most robust and stable solutions with only a single-shot simulation, as well as the best performance overall compared to the benchmarked PIX2PIX with the UNet-256 generator and the state-of-the-art FCNVMB architecture, designed specifically for velocity model building. Our proposed methodology is suitable for training DLI solutions, such as in [33,40], and for closing the existing gaps in the literature for improving DLI. The extension of our methodology to 3D modelling as well as the introduction of features for geological faults will be included in future works, where we will test the method on more complex datasets such as the Marmousi model.

**Author Contributions:** Conceptualisation, A.P. and L.S.; methodology, A.P.; software, A.P.; validation, A.P.; formal analysis, A.P., V.S. and L.S.; investigation, A.P.; resources, A.P.; data curation, A.P.; writing—original draft preparation, A.P.; writing—review and editing, A.P., L.S. and V.S.; visualisation, A.P.; supervision, L.S. and V.S. All authors have read and agreed to the published version of the manuscript.

**Funding:** This research received no external funding.

**Data Availability Statement:** The datasets generated by this paper are publicly available under a CC-BY 4.0 licence on the University of Strathclyde repository at DOI: https://doi.org/10.15129/d49bcfc6-7bd0-450c-9734-cf89403ef9c0, accessed on 12 April 2023.

**Acknowledgments:** We would like to thank Stella Pytharouli for her support and mentorship.

**Conflicts of Interest:** The authors declare no conflict of interest.

## Abbreviations

The following abbreviations are used in this manuscript:

| | |
|---|---|
| FWI | Full Waveform Inversion |
| VMB | Velocity Model Building |
| DNN | Deep Neural Network |
| DLI | Deep Learning- based Inversion |
| RNN | Recurrent Neural Network |
| CNN | Convolutional Neural Network |
| LSTM | Long Short-Term Memory |
| GRU | Gated Recurrent Unit |
| GPU | Graphical Processing Unit |
| MAE | Mean Absolute Error |
| MSE | Mean Squared Error |
| PSNR | Peak Signal to Noise Ratio |
| SSIM | Structural Similarity Index |
| MSSIM | Multiscale Structural Similarity Index |
| REL | Relative Error |
| TWT | Two-Way Time |
| PE | Percentage Error |
| cGAN | Conditional Generative Adversarial Network |
| RGB | Red Green Blue |
| SEG | Society of Exploration Geophysicists |

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
