# Peer review of "Synthetic Data Generation for Deep Learning-Based Inversion for Velocity Model Building"

_remotesensing, doi:10.3390/rs15112901_

Round 1

Reviewer 1 Report

Thank you for sharing your research. This was an interesting study and my comments can be found in the attached file. 

Reviewer 2 Report

The paper reads well and I like the topic presented here. 

I only have two minor suggestions. 

1. It would be great if the authors can connect the work presented here with some existing works on the same topic (e.g., Ren et al., 2021, Building Complex Seismic Velocity Models for Deep Learning Inversion, IEEE Access), and most importantly highlight the scientific advance.

2. It is a pity that I do not see a 3D model in this paper. It would be better to extend the current framework to 3D. If not possible, the authors need to clarify the difficulties. The paper I mentioned above, however, include a detailed workflow on generating 3D realistic model. 

Generally fine. 
